# Interplay between chromosomal alterations and gene mutations shapes the evolutionary trajectory of clonal hematopoiesis

Teng Gao [1,2], Ryan Ptashkin[3], Kelly L. Bolton[4], Maria Sirenko [1], Christopher Fong[1], Barbara Spitzer[5], Kamal Menghrajani[4], Juan E. Arango Ossa[1,2,5], Yangyu Zhou[1,2,5], Elsa Bernard [1,2], Max Levine [1,2,5], Juan S. Medina Martinez[1,2,5], Yanming Zhang[6], Sebastià Franch-Expósito [3], Minal Patel[2], Lior Z. Braunstein[7], Daniel Kelly[8], Mariko Yabe[3], Ryma Benayed [3], Nicole M. Caltabellotta[2], John Philip [9], Ederlinda Paraiso[10,11], Simon Mantha[12], David B. Solit [11,13,14], Luis A. Diaz Jr[13,15], Michael F. Berger [3,16], Virginia Klimek[12,17], Ross L. Levine [2,4,11,15], Ahmet Zehir [3], Sean M. Devlin[18] & Elli Papaemmanuil [1,2,5,18 ✉]

Stably acquired mutations in hematopoietic cells represent substrates of selection that may lead to clonal hematopoiesis (CH), a common state in cancer patients that is associated with a heightened risk of leukemia development. Owing to technical and sample size limitations, most CH studies have characterized gene mutations or mosaic chromosomal alterations (mCAs) individually. Here we leverage peripheral blood sequencing data from 32,442 cancer patients to jointly characterize gene mutations ($n = 14{,}789$) and mCAs ($n = 383$) in CH. Recurrent composite genotypes resembling known genetic interactions in leukemia genomes underlie 23% of all detected autosomal alterations, indicating that these selection mechanisms are operative early in clonal evolution. CH with composite genotypes defines a patient group at high risk of leukemia progression (3-year cumulative incidence 14.6%, CI: 7–22%). Multivariable analysis identifies mCA as an independent risk factor for leukemia development (HR = 14, 95% CI: 6–33, $P < 0.001$). Our results suggest that mCA should be considered in conjunction with gene mutations in the surveillance of patients at risk of hematologic neoplasms.

A full list of author affiliations appears at the end of the paper.

Genome instability is a hallmark of aging and cancer[1]. Recent studies show that stably acquired somatic mutations occur across human tissues throughout life, in an age-dependent manner. A subset of these mutations drive the establishment of "benign" macroscopic clones[2–4] and some may lead to overt neoplastic disease. This phenomenon has been best studied in the context of clonal hematopoiesis (CH)[5,6] due to the relative ease of blood sampling from healthy individuals. Large-scale population-based studies have established a detailed understanding of CH prevalence and mutation characteristics, as well as the incidence of clinical sequelae such as hematologic neoplasms[7–12].

Most CH studies focused on gene mutations including single nucleotide substitutions and indels. CH defined by mosaic chromosomal alterations (mCAs), such as amplifications, deletions, and copy-neutral loss of heterozygosity (CNLOH), has also been reported, albeit at a much lower prevalence[13–18]. In both definitions, CH exhibits a strong age-related incidence and shares similar mutational characteristics with hematological cancers. While gene mutations that target established genes in leukemia pathogenesis are postulated to confer an increased fitness of hematopoietic stem and progenitor cells, the mechanisms of selection driving most mCA events are less well understood. Recent studies show that a proportion of mCAs gain selective advantage through interactions with germline alleles[13,17,18] and allude to a similar mechanism for pre-existing somatic mutations. However, the relationship between acquired gene mutations and mCAs in CH has not been systematically investigated. Progress in this regard has been limited by the lack of statistically powered studies that simultaneously map chromosomal alterations and gene mutations in the same sample[7,9,13–17,19] (Supplementary Table 1).

CH is linked to an increased risk of subsequent hematologic disease suggesting that it likely represents an early step in leukemic transformation. However, most CH cases do not progress to overt disease. Putative driver (PD) status, mutation burden and clone size as estimated by variant allele fraction (VAF) of acquired gene mutations are established predictors of disease progression in CH[10,11]. Such criteria are increasingly being considered clinically to evaluate individuals at risk of leukemic transformation[20–23]. Small-scale retrospective studies have demonstrated that mCAs can either be simultaneously detected with gene mutations or represent the sole detectable abnormalities years before leukemia development[10,24]. However, the clinical significance of mCAs in the context of co-occurring gene mutations has not been investigated.

Recent studies showed that solid tumor cancer patients have high rates of CH and are at an increased risk of secondary leukemias[25–27]. Here we seek to understand the relationship between mCAs, acquired gene mutations, and the incidence of hematological neoplasms in patients with cancer. We develop a method to reliably detect mCAs in data from established prospective sequencing assays used in routine clinical practice. We apply this method in a cohort of 32,442 cancer patients and characterize in detail the incidence and presentation of mCAs in the context of concurrent gene mutations. We provide evidence of direct positive selection mediated by synergistic effects between gene mutations and chromosomal alterations. We show that mCAs are independent predictors of progression to hematological neoplasms and demonstrate the value of incorporating mCAs in the reporting and interpretation of CH in cancer patients[22,23].

## Results

**Patient cohort and samples**. The study population included 32,442 solid tumor patients (median age: 61, range 20–99,

Supplementary Table 2) with non-hematologic cancers that underwent matched tumor and blood sequencing at MSKCC using the MSK-IMPACT panel[28] on an institutional prospective sequencing protocol (ClinicalTrials.gov number, NCT01775072) before February 1, 2020 (Supplementary Fig. 1). The majority of our samples were sequenced by the most recent version of the panel, IMPACT6, which captures the exonic regions of 468 recurrent cancer genes and includes probes against 1042 common single nucleotide polymorphisms (SNPs). This effectively mimics a low-density SNP tiling array with markers evenly distributed across the genome. Clinical annotations including age, gender, race, primary cancer diagnosis, length of follow-up, serial blood counts, and smoking history were available for most patients in the study. For 10,375 patients complete treatment data was previously collected[26]. The patients in this study had no record of hematologic malignancies within 3 months following sequencing (Methods).

**Landscape of mCAs in prospective sequencing of cancer patients**. To study the prevalence of mCAs in patients who have undergone MSK-IMPACT testing, we developed a method, FACETS-CH (a modification of FACETS[29]) that is optimized for detecting chromosomal alterations at low cell fractions from deep targeted sequencing data. The precision of this approach was validated in 919 technical replicates (Methods, Supplementary Figs. 2 and 3). We applied this method to map chromosomal alterations in the matched blood of 32,442 solid tumor patients in our study. We detected 383 mCA events (335 autosomal) in 346 individuals. The incidence of mCA increases with age, from under 1% among individuals under 50 to more than 3% among subjects who were above 80 (Fig. 1a). We used the magnitude of deviations in logR and logOR (Methods) to estimate the aberrant cell fraction and classify events based on copy number states (Fig. 1b); 156 (47%) autosomal events were classified as deletions, 123 (37%) as CNLOH, and 56 (17%) as gains. Among mCA individuals, 317 (92%) harbored one event, whereas 29 (8%) had multiple (range 1–4) events (Fig. 1c). The smallest clone that we detected had an estimated cell fraction of 10% (Fig. 1d), a limit of detection comparable to prior studies based on high-density SNP-arrays without haplotype phasing (Supplementary Fig. 4)[14–16]. The genomic distribution and patterns of the mCAs we detected using a targeted sequencing platform with sparse genome-wide coverage were broadly consistent with findings from high-density SNP array studies, confirming the specificity of our approach[13–16] (Fig. 1e and Supplementary Figs. 5–8). Overall, patients with mCA showed a similar blood count distribution and cytopenia incidence as ones without mCA (Supplementary Fig. 9).

Among different cancer types in our study, mCA was prevalent in soft tissue sarcoma, thyroid, and lung cancer patients and underrepresented in prostate and bladder cancer patients (Fig. 1f). In a multivariable regression model, mCA was significantly associated with age (OR = 1.8, $P < 0.001$), male gender (OR = 1.3, $P = 0.012$), and white race (OR = 1.5, $P = 0.033$), but not smoking (OR = 1.2, $P = 0.18$), which was consistent with findings from prior studies[16] (Supplementary Fig. 10). For a subset of 10,375 patients whose complete clinical treatment history had been previously collected[26], we explored the association of mCA with receipt of oncologic therapy. We found that mCA was positively associated with external beam radiation therapy (OR = 1.7, $P = 0.022$) but not cytotoxic chemotherapy (OR = 0.9, $P = 0.56$) (Supplementary Fig. 10). It is likely that differences in exposures to oncologic therapies may partially account for the distinct mCA representation in different tumor types. However, the association between therapies and mCAs is likely to be heterogeneous[26]. Larger datasets with more complete treatment

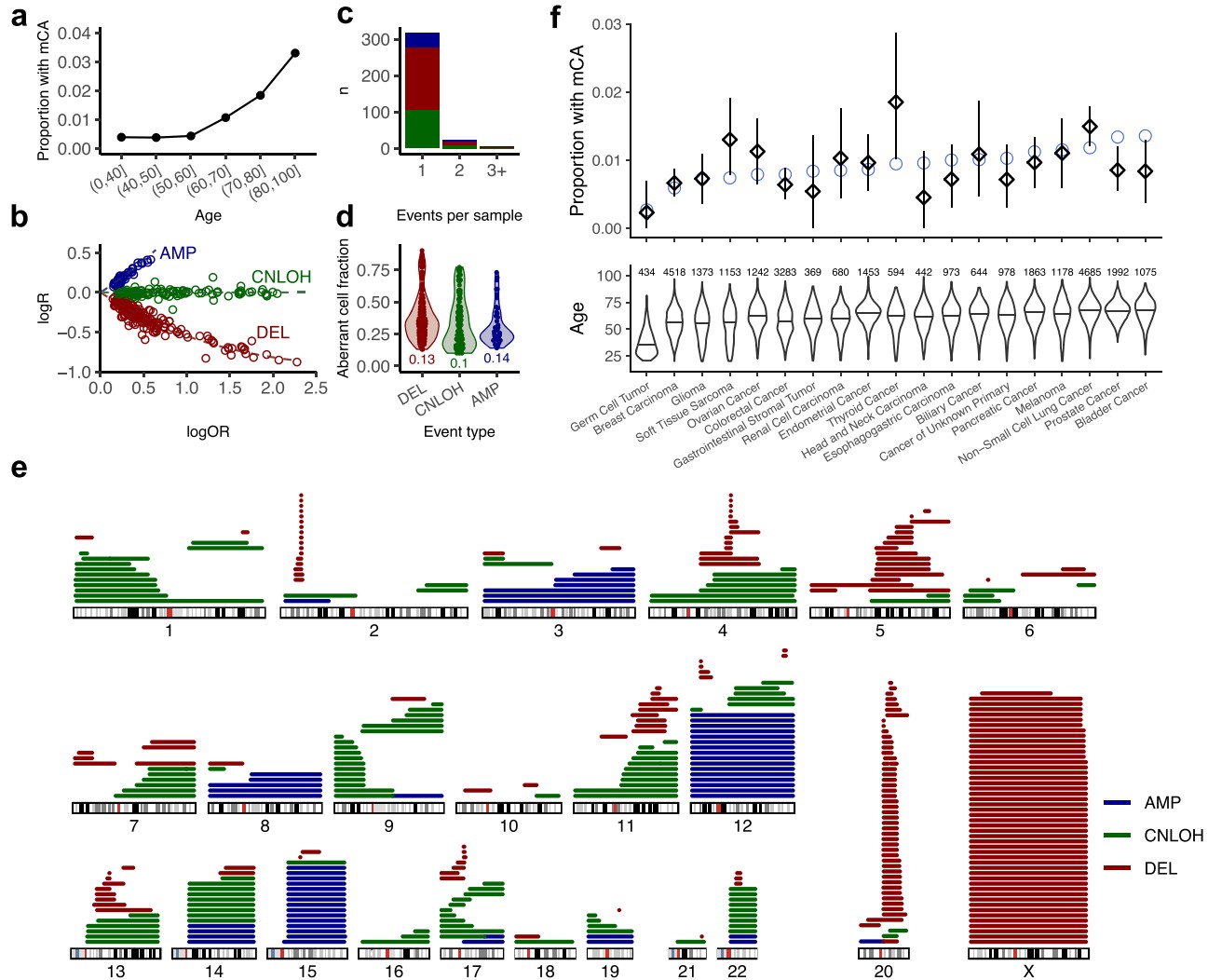

**Fig. 1 Landscape of mCA in the MSK-IMPACT cohort. a** Proportion of subjects with detected mCA increases with age. **b** Classification of events by copy number states. logR: $\log_2$ ratio of the coverage depth between analyzed sample and normal comparison. logOR: log allelic ratio between major and minor alleles in heterozygous SNP loci. Events are colored by inferred alteration type. **c** Number of subjects with 1, 2, or 3+ mCAs. **d** Cell fraction of detected aberrant clones stratified by alteration type. Numbers below the violins indicate the minimum. **e** Genome-wide distributions of detected mCAs. **f** Proportion of subjects with mCA among major cancer types. Squares: observed incidence. Blue circles: expected incidence. 95% CI are shown in black. Number of patients are displayed in the middle. Source data are provided as a Source Data file.

annotation are needed to fully assess the association between specific subclasses of therapy and specific types of mCAs.

**Global characteristics of mCAs in relation to gene mutations.** To evaluate the relationship between mCAs and acquired gene mutations, we characterized gene mutations in our patient cohort using an established variant calling procedure from our previous studies[25,26]. A total of 14,789 mutations were detected in 9854 (30%) individuals across 457 genes captured in our clinical sequencing assay[28] (Supplementary Fig. 11a, b). The incidence of gene mutations increased from 5% among patients in their 20s to more than 60% in elderly patients aged above 80 (Supplementary Fig. 12b). Among the mutation-positive patients, 6919 (70%) patients harbored a single mutation while 2935 (30%) harbored 2 or more (Supplementary Fig. 11c). The estimated clone sizes (median 9%) of these mutations were lower than mCAs detected in this study, which reflects our higher detection sensitivity (VAF ≥ 2%) for gene mutations as compared to mCAs at low cell fractions (Supplementary Fig. 12a). The estimates of mutation prevalence are dependent on the detection sensitivity and

genomic coverage specific to the sequencing assay used in this study as well as characteristics of the MSK-IMPACT patient cohort.

Considering both variant classes, 217 (63%) CH cases with mCAs co-occurred with at least one gene mutation captured by our panel, while 129 (37%) did not (Fig. 2a; Fisher's exact test, OR = 3.9, $P < 0.001$). mCA was especially enriched in CH cases with high mutation number and VAF (Fig. 2b, c). The age distributions for the two types of CH when found in isolation were comparable while the acquisition of both shifted towards older age (Fig. 2d). However, the high degree of overlap between subjects with mCA and gene mutation could not be simply explained by a shared age-related incidence ($P < 0.001$, regression adjusted for age). Taken together, the high degree of co-occurrence of chromosomal aberrations and gene mutations points towards a potential synergistic relationship. This would be consistent with a multi-hit process of carcinogenesis, where the first hit provides a fertile ground for the acquisition of subsequent hits that leads to further selection and expansion of mutant clones.

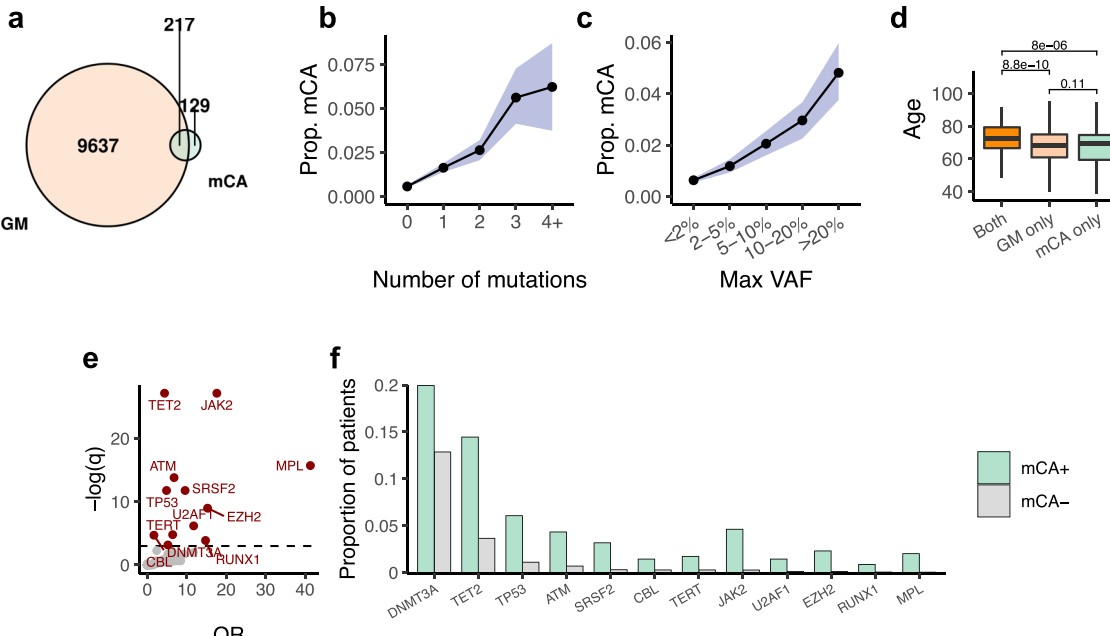

**Fig. 2 Global characteristics of mCA in relation to gene mutations. a** Overlap between CH cases carrying gene mutations (GMs) and chromosomal alterations (mCAs). Number of subjects are shown in black. **b, c** The prevalence of mCA increases with mutation burden and VAF. Observed proportion of mCAs within each subgroup is shown as solid dots; 95% CIs generated from a binomial distribution around the observed proportion are shown in blue shades. Gene mutations in regions affected by mCAs are excluded from this analysis. **d** Age distribution of subjects with mCA only, gene mutations only, and both types of mutations. Unadjusted $P$ values derived from a two-ended Student's $t$-test are shown. The lower and upper bounds of boxes denote 25th (Q1) and 75th (Q3) percentiles of observed ages, respectively. The lower and upper whiskers indicate the minima (Q1 − 1.5*IQR) and maxima (Q3 + 1.5*IQR). **e** Volcano plot of $q$ values and odds ratios of gene representation in mCA-positive versus mCA-negative cases (Fisher's exact test). Genes significantly enriched (FDR < 0.05) in mCA subjects are colored in red. **f** Frequency of gene mutations significantly enriched in subjects with mCA. Source data are provided as a Source Data file.

**Patterns of co-occurrence between mCAs and gene mutations reveal diverse mechanisms of selection.** Population-based studies of leukemia genomes have delivered a detailed understanding of recurrent genetic interactions in overt neoplastic disease[30,31]. Such studies show that chromosomal alterations can be found in *cis* of existing mutations (i.e., overlapping the mutated locus), thereby contributing to an increase in mutant allele dosage, e.g., 9pCNLOH and *JAK2* in myeloproliferative neoplasms (MPN) or bi-allelic inactivation of tumor suppressors, e.g., *TP53* and 17pLOH[32]. Additionally, significant co-occurrences have been reported between specific chromosomal alterations and gene mutations in *trans* (i.e., not overlapping the mutated locus) which suggest functional co-operativity[31,33–37]. Furthermore, the emergence of chromosomal alterations can result from global genomic instability induced by loss of gatekeeper function such as genes (e.g., *TP53*) in the DNA damage response (DDR) pathway[31,32,36]. The enrichment of composite genotypes observed in this study suggests that these genetic interactions may already be in play at the initial stage of somatic evolution in the absence of overt disease.

CH subjects with mCA had a distinct spectrum of gene mutations as compared to the remaining cohort (Fig. 2e, f). Notably, a large proportion of the genes enriched in patients with mCA (*DNMT3A, TET2, TP53, ATM, JAK2, EZH2, RUNX1, MPL*) are also frequently affected by loss of heterozygosity in hematopoietic neoplasms[32,38]. We therefore asked whether the presence of gene mutations is associated with chromosomal alterations in *cis*. Overall, 2% non-coding variants that co-occurred with an mCA event localized in *cis* with the mCA (permutation $P = 0.78$), whereas it is 15% for the coding variants (permutation $P < 0.001$; Supplementary Fig. 13). This enrichment

can be accounted for by mutations in seven genes that recurrently co-localized with an mCA (Fig. 3a, b). Of six events of CNLOH on chr7q, all six co-localized with an *EZH2* (7q36.1) mutation ($q < 0.001$). The high VAFs and the absence of 7qCNLOH events without a concurrent *EZH2* mutation suggest that 7qCNLOH is a highly directed event targeting a previously acquired mutation of *EZH2* in CH (Fig. 3b and Supplementary Fig. 14a). Of 12 cases with 9pCNLOH, 11 (92%, $q < 0.001$) co-localized with a *JAK2* V617F mutation, while 4 out of 9 (44%, $q < 0.001$) 1pCNLOH events co-localized with a *MPL* (1p34.2) mutation. These events mirror *cis* dosage adjustment effects identified in studies of MPN, here identified in patients without active disease. Additionally, we identified recurrent bi-allelic targeting of *TET2* mutations by CNLOH events as well as focal deletions (Fig. 3b). The same is true of chromosomal alterations spanning *DNMT3A* (deletions), *ATM*, and *TP53* (deletions and CNLOH) (Fig. 3b). The cellular fractions of these *cis* multi-hit events generally supported homozygosity of the mutant allele and shared clonal origin (Supplementary Fig. 14a).

We next evaluated co-mutations of mCA and gene mutation in *trans*, which could reflect synergistic effects in cell proliferation or loss of gateway effects promoting global genomic instability[31,32,36]. We scanned for pairwise associations of specific chromosomal alterations with genes recurrently mutated in patients with mCA (Fig. 3c). These analyses revealed highly specific co-mutational patterns reflective of known genetic interactions in a wide range of blood malignancies[30–32,37,39]. *SRSF2/TET2* are commonly co-mutated in myelodysplastic syndromes (MDS) characterized by myelodysplasia and mono-cytosis[40]. In our data, 2 out of 7 (29%) CNLOH and 3 out of 14 (21%) deletions targeting 4q24 co-occurred with mutations in

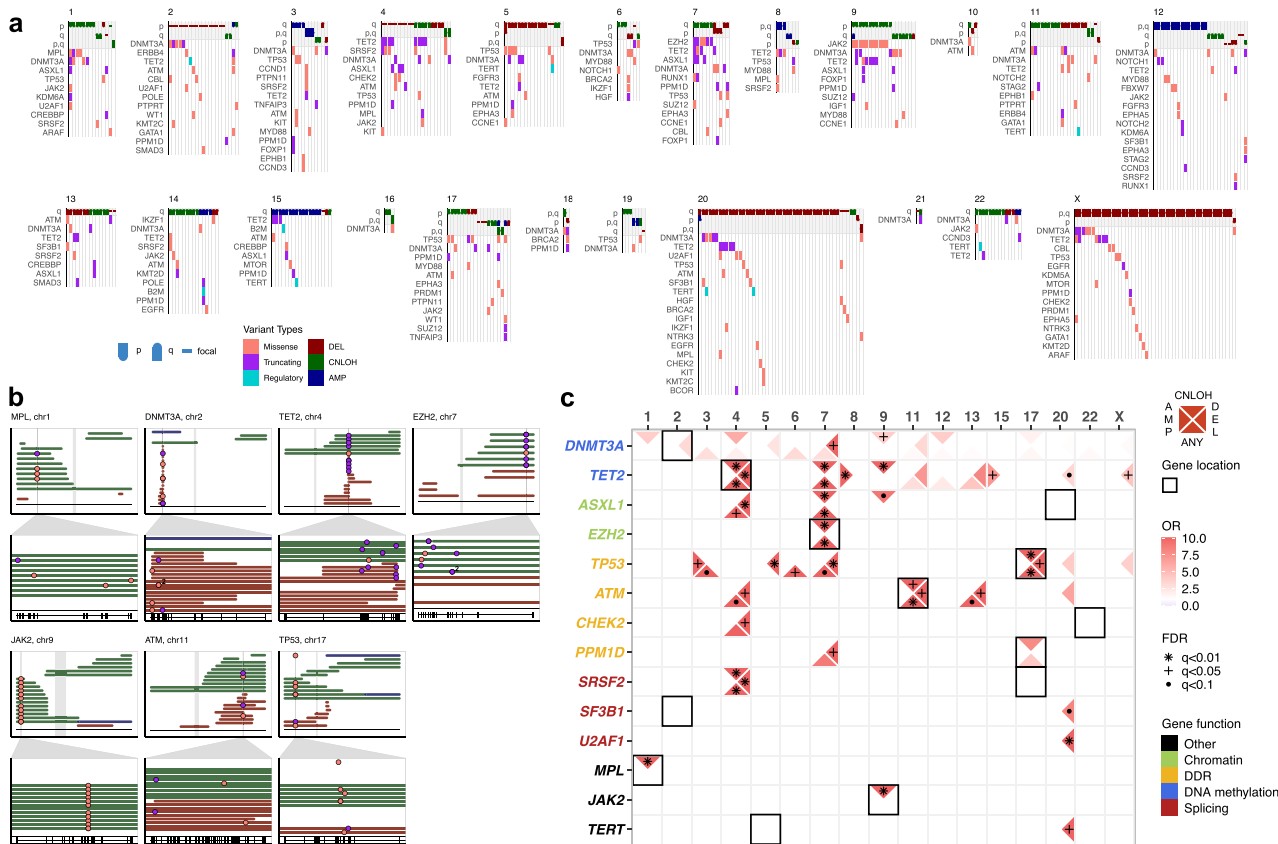

**Fig. 3 Patterns of co-occurrence between mCAs and gene mutations reveal diverse mechanisms of selection. a** mCA events and their co-occurring gene mutations. Events are colored by alteration types. Only genes that appear more than one time are shown. **b** Gene loci that exhibited recurrent double hits in *cis*. Zoom window shows the gene loci affected and the location of the gene mutations; "2" denotes two adjacent mutations in close proximity with each other. Centromere positions are marked in gray. Gene structures are displayed below. **c** Pairwise associations of co-mutations between specific mCAs and gene mutations. Associations with specific chromosomal alteration types are indicated by the position of the triangle (left: AMP, top: CNLOH, right: DEL, bottom: any). FDR-corrected significance values from pairwise Fisher's exact tests are indicated by shapes of asterisks. Odds ratios are indicated by the transparency of red shading. The chromosomal location of genes are indicated by black squares. Gene labels are colored by functional groups. Source data are provided as a Source Data file.

*SRSF2.* Similarly, 3 out of 6 (50%) CNLOH on chr7q co-occurred with a mutation in *ASXL1*, while deletions on chr7 also recurrently co-mutated with *PPM1D*. Mutations in *U2AF1*, albeit rare, co-occurred with 3 out of 44 (7%) cases of 20q deletions (Fig. 3c). These combinations have been previously reported as significantly recurrent in myeloid disease suggesting that they are intrinsically implicated in myeloid pathogenesis[33,41]. Similarly, we also observe co-mutations characteristic of chronic lymphocytic leukemia (CLL); 6 out of 16 (37.5%) cases of trisomy 12 co-occurred with a mutation in *NOTCH1*, *MYD88*, or *FBXW7*; 2 out of 9 (22%) deletions on chr13q co-occurred with a *ATM* mutation[34]. Last, a key part of the DNA damage response pathway, *TP53* acts as a gatekeeper for genomic instability in a wide range of cancers. We found that CH mutations in *TP53* were associated with numerous chromosomal alterations (deletions in chr5, chr7, and trisomy chr3) (Fig. 3c). In addition, we observe a significant association of *TP53* mutations with the presence of multiple concomitant mCAs in CH patients (Fisher's exact test, $P = 0.0015$, Supplementary Fig. 15).

Taken together, acquired gene mutations in *cis* formed putative "double hits" with 26% of all CNLOH events and 8% of deletions in the study. Recurrent co-mutations in *trans* and loss of gatekeeper function (i.e., *TP53*) underlied an additional 13% of deletions and 20% of amplifications. In total, 23% of all autosomal mCA events mapped in this study are subject to these recurrent

co-mutations. Interestingly, most of the subjects with recurrent CH co-mutations did not exhibit significant blood abnormalities at the time of blood draw (Supplementary Fig. 16). It is also of note that certain mCAs (e.g., chr20-, X-) frequently occurred in isolation without any clear co-mutations, indicating that these events can also drive clonal expansions alone (Fig. 3a). While the mechanisms driving CH are becoming increasingly understood for each respective class of mutations, here by integrating chromosomal alterations with gene mutations we unravel previously under-characterized patterns of mutation acquisition reflective of diverse mechanisms of selection operating in CH.

**Evolution to hematologic malignancies**. The discovery of composite genotypes characteristic of hematologic neoplasms in patients without an active disease raises the possibility that they represent markers of incipient leukemic transformation. We reviewed hospital billing records and tumor registry data and identified 96 patients with a confirmed hematological neoplasm diagnosis dated more than 3 months after blood collection (Methods, Supplementary Fig. 17). Of these patients (median time to diagnosis 17 months, range 3–62 months) 26 were diagnosed with MDS, 11 with acute myeloid leukemia (AML), 7 with MPN, 4 with chronic myeloid leukemia (CML), 12 with CLL, 27 with B-cell non-Hodgkin's lymphoma (B-NHL), 2 with

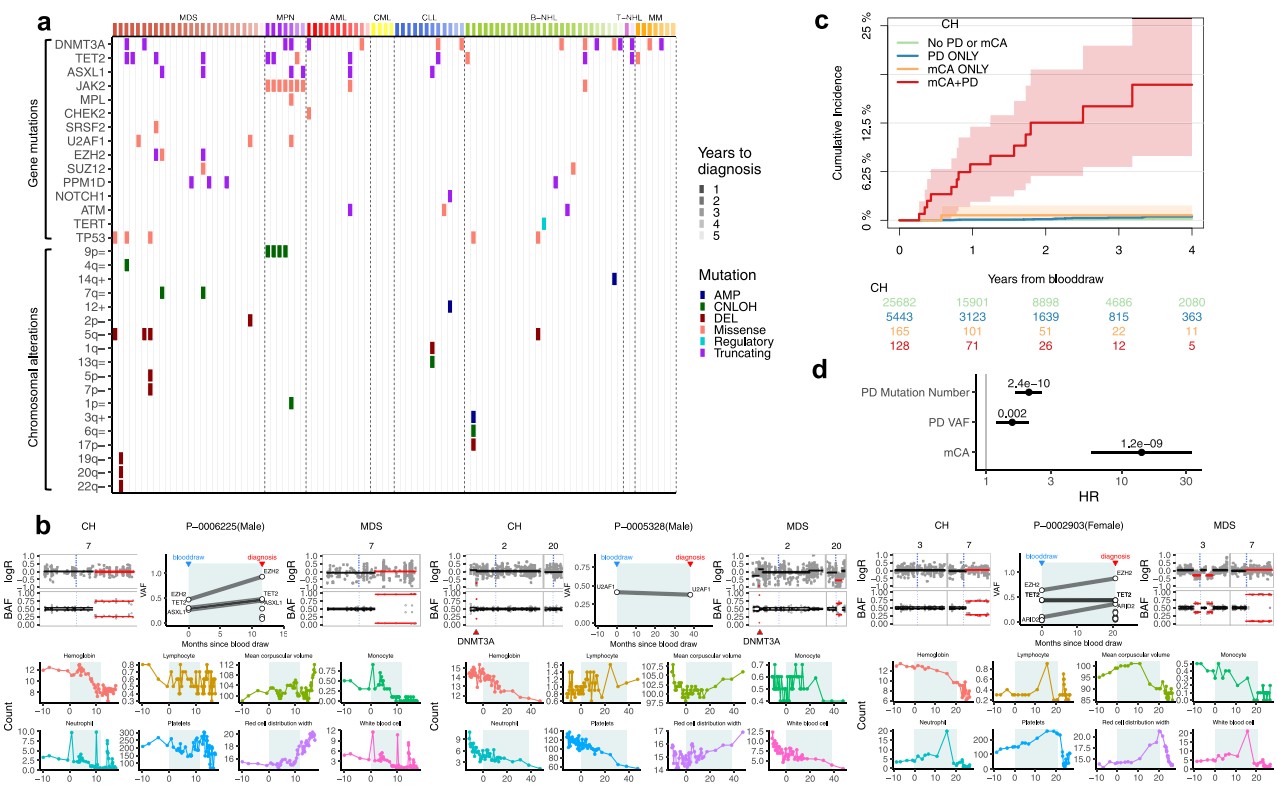

**Fig. 4 mCA and evolution to hematologic malignancies. a** CH mutations detected at initial blood draw in patients who had a subsequent hematological cancer diagnosis. Patients in each disease category (top bar) were arranged by increasing time to diagnosis. Events are colored by mutation types (=, CNLOH; −, deletion; +, amplification). **b** Exemplar cases of genomic evolution from CH with detectable mCA to myeloid neoplasms. Gene mutations at the time points of blood draw and diagnosis are shown in the middle, while regions of chromosomal alterations are shown on the left and right, respectively. Solid lines indicate signal median. Aberrant regions are colored in red. BAF, B-allele frequency. Serial blood count indices are shown below, where the shaded areas in blue indicate the period between initial blood collection and MN diagnosis. **c** Cumulative incidence of leukemia among patients who had detectable mCA only, gene mutation putative driver (PD) only, both, or neither; 95% CIs are shown in shaded ribbons. **d** Association of CH features with subsequent leukemia diagnosis. Hazard ratios (HR; solid dots), 95% CIs (horizontal bars), and unadjusted P values (above horizontal bars) are derived from a multivariable cause-specific Cox regression model with a 9 month landmark. The HR for 10% increment in gene mutation PD VAF was shown. mCA status was included as a binary indicator. Source data are provided as a Source Data file.

T-cell non-Hodgkin's lymphoma (T-NHL), and 7 with multiple myeloma (MM) (Fig. 4a and Supplementary Fig. 18).

Among the 96 patients who received a diagnosis, 40 (42%) had evidence of only gene mutation at the time of blood profiling, 17 (18%) had both gene mutation and mCA, whereas 1 (1%) had mCA only (Fig. 4a). Overall, the mutation patterns observed at blood draw were characteristic of the respective disease types for patients who subsequently developed MDS and MPN, but less so for the remaining disease subsets that are frequently characterized by genomic rearrangements (MM, B-NHL, T-NHL). Among the patients who subsequently developed MDS, those with mCA appeared to have a shorter time to diagnosis (median time to transformation 9.7 months, range: 4.2–38.7) as compared to patients without mCA (median time 23.2 months, range 9–61.8). Likewise, of the 7 patients who developed MPN, the first 5 to be diagnosed (within 22 months) had concomitant *JAK2*/9pCNLOH or *MPL*/1pCNLOH events. This is consistent with studies linking *JAK2* homozygosity to increased disease severity in MPN[42]. In some cases of lymphoma and MM, gene mutations in *DNMT3A* or *TET2* were the only detected events (Fig. 4a). It is unclear whether these mutations were related to the subsequent diagnosis and we therefore restricted our subsequent analysis on the risk of leukemias.

To ascertain the relationship between the mCA clone at initial blood draw and the clone at leukemic transformation, we analyzed leukemia samples from 16 patients that we also sequenced at the time of diagnosis by a comparable targeted panel (Methods). Among the 5 patients with detectable mCA, both gene mutations and mCAs coincided in the same clone that seeded the corresponding hematologic neoplasms (Fig. 4b and Supplementary Fig. 19). In all the cases these clones acquired additional alterations prior to disease transformation.

Aberrant blood counts are a common early symptom of hematologic neoplasms. However, cytopenias are frequent in cancer patients undergoing therapy. Formal hematological evaluations are typically delayed until cytopenias have proven to be persistent after therapy cessation. Serial blood counts in 4 of the cases that subsequently developed MDS (Fig. 4b and Supplementary Fig. 19) showed that composite CH genotypes characteristic of leukemic transformation can be detected 9–39 months prior to diagnosis and before the onset of clinical-grade cytopenia[43].

**Implications for risk stratification**. CH mutational features obtained by NGS assays are increasingly being interpreted in the

clinic to evaluate a patient's risk of leukemic transformation[20,21]. Established criteria are mainly based on gene mutations and involve PD status, variant allele frequency, and number of mutations[10,11,22,23,26]. Although it has been shown that certain mCA events pose an increased risk of subsequent blood cancers, it is unclear whether mCA status adds independent value to the stratification of patients at risk of disease progression when considered together with gene mutations[13,17,18].

During patient follow-up, the 3-year cumulative incidence of leukemias (MDS, MPN, AML, CML, CLL) was significantly higher in patients with composite CH genotypes (14.6%, CI: 7–22%) as compared to patients with either mCA, gene mutation PD only, or neither, of which all had a 3-year cumulative incidence of less than 1% (Fig. 4c). We performed a multivariable cause-specific Cox regression analysis and showed that mCA was independently predictive of subsequent (>9 months post-sequencing) leukemia diagnosis (HR = 14, 95% CI: 6–33, $P = 1.2e-09$) after adjusting for number of mutations and VAF in PDs (Fig. 4d). Importantly, this effect is not restricted to the subset of patients that rapidly progress to overt disease. mCA remains a statistically significant predictor among solid tumor patients who survived and did not yet experience leukemic progression in the course of their clinical care at the time of different landmarks (3, 6, 9, and 12 months post-sequencing; Supplementary Fig. 20). In addition, we obtain similar risk estimates for gene mutations and mCA after restricting the analysis to patients with no blood abnormalities at the time of CH assessment (Supplementary Fig. 21). This demonstrates that incorporating features of chromosomal alterations into the clinical evaluation of CH has the potential to deliver better risk stratification as well as improve the diagnostic workup of patients with suspected hematologic disease.

## Discussion

Recent characterizations of somatic mutations in cancer as well as normal tissues suggest that the multi-step process of tumor-igenesis is often an interplay between gene mutations and chromosomal alterations[2,3,44]. Studies of CH provide an opportunity to elucidate the mechanisms of mutation acquisition, clonal selection, and malignant transformation. Owing to limitations in genotyping platforms and sample size, most CH studies have largely been restricted to either gene mutations or mCAs (Supplementary Table 1)[7,9,13–17,19]. Consequently, such studies do not permit full characterization of the dynamic processes underlying the early steps of clonal evolution. Here, we present a large-scale integrative analysis of mCAs and gene mutations in CH. We leveraged data from a prospective sequencing cohort of 32,442 cancer patients profiled by high-depth targeted capture sequencing of 468 cancer genes and genome-wide copy number probes (MSK-IMPACT) to profile the landscape of mCAs in conjunction with gene mutations. Since our methodology is biased towards the detection of mCAs in relatively large clones (≥10%), it is likely that mCAs that are preferentially present in low cell fractions or in regions not optimally tiled by our panel may have been missed (Supplementary Figs. 5 and 6). Nonetheless, we were able to detect events with comparable sensitivity to prior studies based on unphased high-density SNP-arrays and recapitulate the genomic distribution, age-related incidence, and established demographical associations of mCAs[14–16]. In addition, our more sensitive approach to detect gene mutations (≥2% VAF) in the same sample relative to prior studies[7,9,18,19] enables more comprehensive characterization of the co-mutational landscape of mCAs that we detect.

Although the selection forces driving the clonal expansion of somatic gene mutations are relatively well understood, the mechanisms underpinning the expansion of most mCAs remain

unclear[13,18,45]. Recent studies have shown that at least 18% of CNLOH events can be attributed to germline alleles[13,17,18]. However, it is likely that the selection mechanisms of mCAs are multifactorial. Here we aim to uncover factors affecting the acquisition of mCAs in the context of demographic variables, environmental exposures, as well as CH gene mutations. We recapitulate known associations of mCA with age (OR = 1.8, $P <$ 0.001), male gender (OR = 1.3, $P = 0.012$), and white race (OR = 1.5, $P = 0.033$)[14–16]. We find that mCAs are significantly associated with external beam radiation (OR = 1.7, $P = 0.022$), but not cytotoxic chemotherapy (OR = 0.9, $P = 0.56$). This result is in line with the highly heterogeneous effects of cancer therapy on CH shown in recent studies[25,26]. It is possible that associations may exist between specific types of mCAs and specific classes of cytotoxic therapy, but delineation of such effects will require larger datasets with more detailed treatment annotations.

In the present study, we analyze mCA presentation in the context of CH gene mutations and provide evidence that mCAs gain selective advantage through complex patterns of co-mutations. Approximately two-thirds of mCA cases had a concurrent gene mutation in one of the cancer-related genes in our panel. Out of the 57 coding gene mutations that co-localized with a deletion or CNLOH event in *cis*, 49 (86%) mapped in one of the 7 genes (*DNMT3A, TET2, JAK2, MPL, EZH2, TP53,* and *ATM*). This implies a gain of fitness that is mediated by bi-allelic alterations, resulting in either oncogene mutant dosage adjustment or inactivation of tumor suppressors. Certain mCA events, such as 7qCNLOH, 4/4qCNLOH, 9pCNLOH, and 1pCNLOH, were highly directed events acting on previously acquired gene mutations in the corresponding loci (*EZH2, TET2, JAK2,* and *MPL*). Together, these recurrent *cis* co-mutations underlied 26% of CNLOH events and 13% of all mCA events in this study. In addition, we observe recurrent composite genotypes indicative of co-operating or epistatic interactions as well as loss of gatekeeper function (i.e., *TP53*) presenting with multiple chromosomal aneuploidies. These co-mutations in *trans* underlied at least an additional 9.6% of mCA events. Taken together, our results reveal a previously under-characterized layer of complexity in the evolutionary dynamics of CH that converges towards characteristic genotypes associated with distinct leukemia subtypes. Analysis of paired samples at the time of transformation further demonstrated the co-presence of composite genotypes within the same precursor clone that further evolved to seed the subsequent leukemia. This puts mCAs in the context of the continuous evolutionary process of oncogenesis that can often span years and sheds lights on its patterns of acquisition and progression. Future developments of technology platforms that can reliably map gene mutations and chromosomal aberrations jointly at a single-cell resolution are warranted to fully resolve their clonal relationships.

Screening of CH mutations by NGS assays has been increasingly recognized as a potential strategy to identify individuals at risk of hematological neoplasms. This is particularly relevant for cancer patients, who are at a heightened risk of secondary leukemia. Existing screening approaches center around the gene-mutation features of CH including PD status, mutation number, and VAF. By multivariable analysis we show that mCA is an independent risk factor of leukemia in cancer patients. The mCA events that we characterize in this study are present in a relatively large cell fraction (≥10%), which means that they have already undergone substantial clonal expansion. It is likely that the selection forces and co-mutational patterns shaping the spectrum of smaller mCA clones are different. Similarly, interpretation of the clinical relevance of smaller mCA clones and their co-mutations will require more sensitive detection methods for subclonal alterations. It is also of note that due to the relatively short follow-up, we are limited in our capacity in assessing the

long-term effects of mCAs on leukemia risk, which would be a goal for future studies. Nonetheless, our study demonstrates that the integration of chromosomal aberrations provides additional resolution to risk stratification as well as interpretation of clinical phenotypes that are often confounded by oncologic therapy (i.e., cytopenias), and that mCAs should be screened in conjunction with gene mutations. Beyond individuals at risk, our data support that there is an opportunity to also identify individuals with suspected hematologic disease that are difficult to capture with current clinical surveillance. To this end, we deliver a computational framework that enables the simultaneous characterization of mCA and gene mutation from a single assay. Fortuitously, solid tumor patients increasingly undergo routine sequencing of not only tumor biopsies, but also matched blood samples as germline controls[28,46] thus enabling comprehensive CH surveillance programs to be readily integrated in routine cancer care.

## Methods

**MSK-IMPACT cohort.** The study population included patients with non-hematologic cancers at MSKCC that underwent matched tumor and blood sequencing using the MSK-IMPACT panel on an institutional prospective tumor sequencing protocol ClinicalTrials.gov number, NCT01775072 before February 1, 2020 (Supplementary Fig. 1); all patients enrolled on this protocol provided informed consent in written form. The analysis plan, sample collection, and data usage were approved by the Institutional Review Board (IRB) of Memorial Sloan Kettering Cancer Center. A subset of patients that underwent tumor-genomic profiling as standard of care were not directly consented, in which case an IRB waiver was obtained to allow for inclusion into this study. We extracted data on race, smoking, date of birth, and cancer history through the MSK cancer registry and Darwin Digital Portal (DDP). Subjects who had a billing code related to hematologic malignancy prior to, or less than 3 months after the time of blood draw were excluded (Supplementary Fig. 17). Subjects aged less than 20 years were excluded since we included these samples in the unmatched pool of normal. When unavailable through the cancer registry, we extracted data on race and smoking through structured fields in clinician medical notes if available. Blood indices were taken from clinical labs closest to the date of blood collection for MSK-IMPACT, within 15 months before or after blood collection (median = 0 month). Subjects for which age or gender was not available were excluded.

**Targeted capture-based sequencing.** Subjects had a tumor and blood sample (as a matched normal) sequenced using MSK-IMPACT, a FDA-authorized hybridization capture-based next-generation sequencing assay encompassing all protein-coding exons of 341, 410, or 468 cancer-associated genes. MSK-IMPACT is validated and approved for clinical use by the New York State Department of Health Clinical Laboratory Evaluation Program and is used to sequence cancer patients at Memorial Sloan Kettering. Genomic DNA is extracted from deparaffinized formalin-fixed paraffin-embedded (FFPE) tumor tissue and patient-matched blood samples, sheared and DNA fragments were captured using custom probes. All sequencing samples were checked for inter-individual contamination through genotyping of 1042 common SNPs targeted by the sequencing panel.

**Paired leukemia samples.** We sequenced the paired leukemia sample of 16 patients who subsequently developed a myeloid neoplasm (12 MDS, 4 AML) on a comparable gene panel (14 IWG, 1 MSK-HEMEPACT, 1 IMPACT).

**Technical replicate and serial samples.** We had access to a second sequencing sample for 1200 patients, among which 919 are technical replicates (same blood sample sequenced on separate sequencing runs) and 281 are serial samples (different blood samples collected at different dates sequenced on separate sequencing runs).

**Panel of normal samples.** Normal blood sequences of 291, 141, and 22 patients aged <20 years were used as panel of normals for samples sequenced by IMPACT6, IMPACT5, IMPACT3, respectively.

**mCA detection using FACETS-CH.** We modified the FACETS algorithm to detect deeply subclonal chromosomal alterations[29]. We will refer to this algorithm as FACETS-CH. Let logR denote the $\log_2$ ratio of the coverage depth between analyzed sample and normal comparison, which reflects deviations in total copy number from diploid state, while logOR denote the log allelic ratio between major and minor alleles in heterozygous SNP loci, which reflects allelic imbalance. Because no matched normal reference is available, we inferred SNP loci with a B-allele frequency (BAF) between 0.1 and 0.9 to be heterozygous (for this reason, we did not detect clonal deletions or CNLOH). We will refer to the vector of logR as **X**

and the vector of logOR as **Y**. Given a sample, FACETS-CH generates a logR profile compared to each normal $i$ in the PON and selects the logR profile that minimizes noise, which is defined as the square sum of logR of each marker $j$.

$$\epsilon(i) = \sum_{j=1}^{N} X_j^2(i) \tag{1}$$

$$i^* = \underset{i \in 1,\dots,m}{\mathrm{argmin}}\, \epsilon(i) \tag{2}$$

here $N$ is the total number of markers and $m$ the total number of PON samples. We pre-selected stable markers (SNPs positions from DBSNP 138.b37) that show the expected distribution of logR in the PON for each respective sequencing platform. We also profiled the allelic mapping bias for each SNP marker based on the logOR distribution in the PON beforehand. A corrected logOR profile can be generated by subtracting the allelic bias $r$ in the PON from the raw logOR value computed by FACETS.

$$Y_j = \log\left(\frac{a_j}{b_j}\right) - \log\left(r_j\right) \tag{3}$$

Segments are generated by the FACETS CBS segmentation algorithm. We derive a bivariate Wald test statistic for each segment to assign it a probability of being "aberrant" as below. For each segment $s$, we generate all possible contiguous samples of the same number of markers $n$, and calculate standard error of the mean in logR and logOR.

$$SE_X = \sqrt{\frac{\sum_{j=1}^{N-n+1}\left(\bar{X}_{[j,j+n-1]} - \overline{\overline{X}}\right)^2}{N - n}} \tag{4}$$

$$SE_Y = \sqrt{\frac{\sum_{j=1}^{N-n+1}\left(\bar{Y}_{[j,j+n-1]} - \overline{\overline{Y}}\right)^2}{N - n}} \tag{5}$$

In practice, we find an alternate estimation of $SE_Y$ to be more reliable:

$$SE_Y = \frac{\sqrt{\frac{\sum_{j=1}^{N}(Y_j - \bar{Y})^2}{N-1}}}{\sqrt{n}} \tag{6}$$

We then normalize the deviations in logR and logOR signals into Z scores.

$$Z_X = \frac{\bar{X}_s - \mu_X}{SE_X} \xrightarrow{\mathcal{D}} \mathcal{N}(0,1) \tag{7}$$

$$Z_Y = \frac{\bar{Y}_s - \mu_Y}{SE_Y} \xrightarrow{\mathcal{D}} \mathcal{N}(0,1) \tag{8}$$

Assuming that $Z_X$ and $Z_Y$ are approximately independent, the sum of the two squared Z scores gives the desired chi-squared statistic with 2 degrees of freedom. A P value can then be calculated from the statistic using a one-tailed test.

$$T = Z_X^2 + Z_Y^2 \xrightarrow{\mathcal{D}} \chi^2(2) \tag{9}$$

FACETS-CH initializes by assuming no aberrant segment is present in the copy number profile and iterates between two states: (1) profiling noise in normal regions and (2) calling aberrant regions. At the end of each iteration, the aberrant states (denoted by vector **a**) are updated. The stop condition is met if no new segment is assigned as aberrant at the end of the current iteration. Pseudocode of this procedure is shown below.

```
procedure CALLABERRANT(a, X, Y)
    a' ← UPDATEA(a, X, Y)    ▷ Make calls based on noise in normal regions
    if |a'| = |a| then              ▷ Stop if no new aberration is called
        return a
    else                    ▷ Otherwise continue using the new aberrant profile
        return CALLABERRANT(a', X, Y)
```

**Assignment of mCA event types.** Following mCA detection, we classify candidate events as deletion, gain, or CNLOH based on the magnitude of deviations in logR and logOR. Each event type can be defined by the paternal copy number $p$ and maternal copy number $m$: $p = 1$, $m = 0$ for deletion, $p = 2$, $m = 1$ for amplification, and $p = 2$, $m = 0$ for CNLOH. Each genotype $g$, defined by a tuple $(p, m)$, gives rise to an expected logR and logOR deviation for a given cell fraction:

$$X(\phi, p, m) = \log_2\left(\frac{2 \cdot (1 - \phi) + (m + p) \cdot \phi}{2}\right) \tag{10}$$

$$Y(\phi, p, m) = \left|\log\left(\frac{m \cdot \phi + 1 - \phi}{p \cdot \phi + 1 - \phi}\right)\right| \tag{11}$$

We can then calculate the expected logR and logOR for the three event types given any cell fraction from 0 to 1. For each aberrant segment $s$, we define an error score

$\varepsilon$ as the squared Euclidean distance of the observed and expected logR and logOR given a cell fraction and genotype. We find the optimal genotype and cell fraction by minimizing this error score over all possible cell fractions from 0 to 1 and all possible genotypes.

$$\epsilon(g, \phi) = (X(\phi, g) - \bar{X}_s)^2 + (Y(\phi, g) - \bar{Y}_s)^2 \quad (12)$$

$$(g^*, \phi^*) = \operatorname*{argmin}_{g, \phi} \epsilon(g, \phi) \quad (13)$$

**Validation of mCA calling method**. To assess the reproducibility of the method, we applied the same calling procedure to 919 paired replicate samples. Out of 10 total variants that were called, all were reproduced in a separate sequencing run (Supplementary Figs. 2 and 3). We also compared the clonal representation, genomic distribution, and population prevalence of specific mCA events with prior studies (Supplementary Figs. 5–8), which were broadly concordant. The differences in the relative abundance of specific events can be explained by the varying density of available SNPs between genomic regions in a targeted sequencing panel, difference in genotyping platforms, and variant calling sensitivity.

**Gene mutation detection**. Variant calling for each blood sample was performed unmatched, using a pooled control sample of DNA from 10 unrelated individuals as a comparator. Single nucleotide variants (SNVs) were called using Mutect and VarDict. Insertions and deletions were called using Somatic Indel Detector (SID) and VarDict. Variants that were called by two callers were retained. Dinucleotide substitution variants (DNVs) were detected by VarDict and retained if any base overlapped a SNV called by Mutect. All called mutations were genotyped in the patient-matched tumor sample. Variant calls that were present in the blood with a VAF of at least twice that in the tumor or 1.5 times the VAF if the tumor biopsy site was a lymph node were considered somatic. We applied additional post-processing filters in order to eliminate false positive calls (Supplementary Methods). We did not report silent or intronic mutations.

**Gene mutation annotation**. Mutations were annotated with VEP (version 86) and OncoKb. All variants were annotated for their functional relevance as putative drivers in cancer (CH-PD) or specifically to myeloid neoplasms (CH-Myeloid-PD). We annotated variants as CH-Myeloid-PD if they met any of the following criteria:

1. Truncating variants in *NF1, DNMT3A, TET2, IKZF1, RAD21, WT1, KMT2D, SH2B3, TP53, CEBPA, ASXL1, RUNX1, BCOR, KDM6A, STAG2, PHF6, KMT2C, PPM1D, ATM, ARID1A, ARID2, ASXL2, CHEK2, CREBBP, ETV6, EZH2, FBXW7, MGA, MPL, RB1, SETD2, SUZ12, ZRSR2,* or in *CALR* exon9.
2. Translation start site mutations in *SH2B3*.
3. *TERT* promoter mutations and *FLT3*-ITDs.
4. In-frame indels in *CALR, CEBPA, CHEK2, ETV6, EZH2*.
5. Any variant occurring in the COSMIC "haematopoietic and lymphoid" category greater than or equal to 10 times.
6. Any variant noted as potentially oncogenic in an in-house dataset of 7000 individuals with myeloid neoplasm greater than or equal to 5 times.

We annotated variants as CH-PD if they met any of the following criteria:

1. Any variant noted as oncogenic or likely oncogenic in OncoKB.
2. Any truncating mutations (nonsense, essential splice site, or frameshift indel) in known tumor-suppressor genes as per the Cancer Gene Census or OncoKB.
3. Any variant reported as somatic at least 20 times in COSMIC.
4. Any variant meeting criteria for CH-Myeloid-PD as above.

**Hematological cancer diagnosis**. We used medical coding data to identify patients who received a hematological malignancy diagnosis (Supplementary Figs. 1 and 17). This includes ICD-9/ICD-10 codes from episodic diagnosis coding that were generated for each patient encounter at MSK, and ICD-O codes from tumor registry coding based on a review of an entire patient chart 6 months after initial diagnosis. A date cutoff is applied to identify patients who had an active hematological diagnosis prior to or within 3 months of blood collection, whom we excluded from all analyses. We used medical codes received more than 3 months after initial blood collection to identify patients who subsequently developed a hematological cancer during the follow-up period.

**Expected mCA frequency among cancer types**. We built a logistic regression model using the full dataset, with the presence of autosomal mCA as the outcome and age, gender, and race as covariates. The predicted probability of having an mCA event for each individual was averaged to produce an expected frequency for each cancer type.

**Risk factors of mCA**. We explored the associations of mCA with demographic factors and tobacco smoking among 30,870 individuals with complete

demographical data. We performed a logistic regression with the presence of autosomal mCA as the outcome against gender, race (white vs other), age in decades, and smoking status (ever smoker) as covariates. We then explored the association of mCA with oncologic therapy in 10,340 patients with complete clinical treatment history and demographical data. Here we performed a second logistic regression with the presence of autosomal mCA as the outcome against ever receiving XRT and ever receiving cytotoxic chemotherapy by the time of blood collection, adjusted for gender, race, age, and smoking status.

**Rescue of high-VAF mutations in regions affected by LOH**. We conducted a systematic review of gene loci that overlapped a deletion or CNLOH event to identify any homozygous mutations at high VAF (>35%) that would have otherwise been missed by our current set of filters. In order to ascertain the somatic origin of candidate mutations, we compared their VAF in the blood versus the tumor, taking into consideration potential blood infiltration in the tumor sample. We additionally checked for LOH overlapping the mutation locus in the tumor sample to rule out heterozygous SNPs that are potentially lost in comparison. Using this approach, we rescued 14 mutations in *cis* with a chromosomal alteration, including 7 in *TET2*, 4 in *EZH2*, 1 in *MPL*, 1 in *TP53*, and 1 in *JAK2*.

**Enrichment of gene mutations in subjects with mCA**. To identify gene mutations associated with mCA, we conducted a systematic scan of all commonly mutated genes using a two-tailed Fisher's exact test. We restricted our analysis to 138 genes mutated more than 30 times in CH subjects. To control false discovery rate, a q value was calculated for each association using the Benjamini-Hochberg procedure.

**Enrichment of *cis* configuration**. To estimate the expected proportion of *cis* versus *trans* configurations among co-occurring gene mutations and chromosomal alterations, we randomly shuffled original samples of gene mutations that co-occurred with any autosomal mCA and recalculated the proportion of *cis* configuration after each shuffle. We performed 2000 permutations in total. The upper and lower confidence interval of the expected *cis* portions were calculated by the 95th and 5th quantile of the empirical distribution. Similarly, a one-tailed *P* value for the observed *cis* proportions was derived from the empirical distribution.

**Genetic interactions in *cis***. To detect mCAs that target specific gene mutations, we scanned for associations between chromosomal alterations overlapping a gene locus (or within a window of 5MB) and gene mutations in that gene using a two-tailed Fisher's exact test. We restricted our analysis to CNLOH and deletions overlapping the top 14 mutated genes among mCA subjects (*DNMT3A, TET2, ASXL1, EZH2, TP53, ATM, CHEK2, PPM1D, SRSF2, SF3B1, U2AF1, MPL, JAK2, TERT*), yielding 2 * 14 = 28 total tests. To control false discovery rate, a q value was calculated for each association using the Benjamini-Hochberg procedure. We only reported event pairs that co-occurred at least 2 times.

**Genetic interactions in *trans***. To identify gene mutations significantly co-mutated with specific mCAs, we performed a pairwise two-tailed Fisher's exact test between commonly mutated genes and mCAs. We categorized mCA events by chromosome and alteration type including "deletions", "CNLOH", "amplification", or "any". If >70% of the events on a chromosome belong to a single alteration type (e.g., chr20), we excluded the "any" category for that chromosome. We restricted our analysis to mCA events that had at least 4 carriers and the top 14 genes mutated among mCA subjects, yielding 38 * 14 = 532 combinations. To control false discovery rate, a q value was calculated for each pairwise association using the Benjamini-Hochberg procedure. Since the tests for "any" category (*n* = 154) are not independent of those for specific alteration types, we penalized those tests separately from the rest (*n* = 378). We only reported event pairs that co-occurred at least 2 times.

**Risk of hematological malignancies**. To evaluate the association between CH features (including PD mutation VAF, PD mutation number, and presence of mCA) and the risk of subsequent hematological malignancies, we performed cause-specific Cox proportional hazards regression adjusted for age. Patients alive without hematological malignancies were censored at the latest date of contact. With each chosen landmark, all patients who either received a diagnosis or were censored before the landmark were excluded from the analysis, and the start of follow-up was reset at the landmark.

**Reporting summary**. Further information on research design is available in the Nature Research Reporting Summary linked to this article.

## Data availability

mCA and gene mutation calls of 32,442 patients are available in Supplementary Data 1. NIH SEER hemeDB can be accessed at https://seer.cancer.gov/seertools/hemelymph. Raw sequencing data cannot be publicly deposited for legal and privacy reasons, as sequencing was performed for clinical purposes. Mutation calls will also be available

online on cBioPortal (https://www.cbioportal.org). Patient demographics and clinical data are available on GitHub (https://github.com/papaemmelab/Gao_NC_CH). The remaining data are available within the Article, Supplementary Information, or available from the authors upon request. Source data are provided with this paper.

## Code availability

All data analysis was conducted using R 3.6.1 (https://www.r-project.org/) and JupyterLab 1.1.4 (https://jupyter.org/). A GitHub repository containing the code used in the analysis and the FACETS-CH method is available (https://github.com/papaemmelab/Gao_NC_CH).

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

## Acknowledgements

This work was supported by the National Institute of Health (K08CA241318 to K.L.B., P30 CA008748 to S.M.D., UG1-HL069315 to V.K., F31 CA254130-01 to M.S.), American Society of Hematology (K.L.B. and E. Papaemmanuil), EvansMDS Foundation (K.L.B.), European Hematology Association (E. Papaemmanuil), Gabrielle's Angels Foundation (E. Papaemmanuil), V Foundation (E. Papaemmanuil), Geoffrey Beene Foundation (E. Papaemmanuil), Damon Runyon (E. Papaemmanuil), Josie Robertson (E. Papaemmanuil), Cycle for Survival (V.K.), Starr Cancer Consortium (R.L.L., A.Z., M.F.B., R.P.). The authors would like to acknowledge all members of the Papaemmanuil laboratory, the Center for Heme Malignancies, the Precision Interception Prevention (PIP) initiative, the High Performance Computing (HPC) group, and the Integrative Genomics Operation (IGO) core at MSKCC for providing the resources to conduct this research. The authors would also like to thank Dr. Dan Landau for inspiring and reviewing the manuscript.

## Author contributions

T.G. and E. Papaemmanuil conceived and designed the study, oversaw all method development, data analyses and wrote the manuscript. T.G., C.F., K.L.B., S.M.D., Y. Zhang, J.P., B.S., K.M., L.Z.B., D.K., M.Y., N.M.C., and M.P. performed collection and curation of clinical data. R.P., R.B., M.F.B., A.Z., and D.B.S. led the generation of IMPACT sequencing data. K.L.B., M.P., V.K., R.L.L., S.M.D., and N.M.C. performed collection of sequential samples. T.G., R.P., and E. Papaemmanuil developed the method for chromosomal alteration mapping from IMPACT sequencing data with input from E.B. and M.L.; R.P., T.G., A.Z., and E. Papaemmanuil performed gene mutation calling and post-processing of sequencing data. T.G., S.M.D., K.L.B., M.S., S.F.E., E.B., and E. Papaemmanuil performed statistical analyses and/or participated in data interpretation. J.E.A.O., J.S.M.M., M.L., and Y. Zhou oversaw bioinformatics application development and execution of data generation. L.A.D., E. Paraiso, M.F.B., and R.L.L. oversaw the MSK CH program. All authors reviewed the manuscript and approved it for submission. Authorship contributions were evaluated and approved by the MSKCC CH Data usage committee.

## Competing interests

The authors declare the following competing interests: K.L.B. has received research funding from GRAIL; E.B. receives research funding from Celgene. D.B.S. has served as a consultant/received honoraria from Pfizer, Loxo Oncology, Lilly Oncology, Illumina, and Vivideon Therapeutics. M.F.B has participated in advisory board activities for Roche and has received research support from Grail and Illumina. S.M.D. is principal owner of Daboia Consulting LLC. L.A.D. is a member of the board of directors of Personal Genome Diagnostics (PGDx) and Jounce Therapeutics. He is a paid consultant to PGDx and Neophore. He is an uncompensated consultant for Merck but has received travel and research support for clinical trials from Merck. L.A.D. is an inventor of multiple licensed patents related to technology for circulating tumor DNA analyses and mismatch repair deficiency for diagnosis and therapy from Johns Hopkins University. Some of these licenses and relationships are associated with equity or royalty payments directly to Johns Hopkins and L.A.D. His wife holds equity in Amgen. The terms of all these arrangements are being managed by Johns Hopkins and Memorial Sloan Kettering in accordance with their conflict of interest policies. J.S.M.M. is a member of the board of directors and holds equity in Isabl, a software analytics company for high-throughput clinical whole-genome and RNA-sequencing analyses. R.L.L. is on the supervisory board of Qiagen and is a scientific advisor to Loxo, Imago, C4 Therapeutics, and Isoplexis which include equity interest. He receives research support from and consulted for Celgene and Roche, and has consulted for Lilly, Janssen, Astellas, Morphosys, and Novartis. He has received honoraria from Roche, Lilly, and Amgen for invited lectures and from Gilead for grant reviews. A.Z. received honoraria from Illumina. E. Papaemmanuil receives research funding from Celgene and has received honoraria for speaking and scientific advisory engagements with Celgene, Prime Oncology, Novartis, Illumina, and Kyowa Hakko Kirin. E. Papaemmanuil is also a member of the board of directors and holds equity in Isabl, a software analytics company for high-throughput clinical whole-genome and RNA-sequencing analyses. E. Papaemmanuil is an inventor in software licenses related to technology of genome analytics. Some of these licenses and relationships are associated with equity or royalty payments to MSKCC and are managed accordingly by the conflict of interest office at MSKCC. The remaining authors declare no competing interests.

## Additional information

¹Computational Oncology Service, Department of Epidemiology & Biostatistics, Center for Computational Oncology, Memorial Sloan Kettering Cancer Center, 1275 York Ave, New York, NY 10065, USA. ²Center for Hematologic Malignancies, Memorial Sloan Kettering Cancer Center, 1275 York Ave, New York, NY 10065, USA. ³Department of Pathology, Memorial Sloan Kettering Cancer Center, 1275 York Ave, New York, NY 10065, USA. ⁴Department of Medicine, Leukemia Service, Memorial Sloan Kettering Cancer Center, 1275 York Ave, New York, NY 10065, USA. ⁵Department of Pediatrics, Memorial Sloan Kettering Cancer Center, 1275 York Ave, New York, NY 10065, USA. ⁶Department of Pathology, Cytogenetics Laboratory, Memorial Sloan Kettering Cancer Center, 1275 York Ave, New York, NY 10065, USA. ⁷Department of Radiation Oncology, Memorial Sloan Kettering Cancer Center, 1275 York Ave, New York, NY 10065, USA. ⁸Department of Information Systems, Memorial Sloan Kettering Cancer Center, 1275 York Ave, New York, NY 10065, USA. ⁹Department of Health Informatics, Memorial Sloan Kettering Cancer Center, 1275 York Ave, New York, NY 10065, USA. ¹⁰Center for Strategy & Innovation, Memorial Sloan Kettering Cancer Center, 1275 York Ave, New York, NY 10065, USA. ¹¹Human Oncology and Pathogenesis Program, Memorial Sloan Kettering Cancer Center, 1275 York Ave, New York, NY 10065, USA. ¹²Department of Medicine, Hematology Service, Memorial Sloan Kettering Cancer Center, 1275 York Ave, New York, NY 10065, USA. ¹³Department of Medicine, Memorial Sloan Kettering Cancer Center, 1275 York Ave, New York, NY 10065, USA. ¹⁴Department of Medicine, Solid Tumor Division, Memorial Sloan Kettering Cancer Center, 1275 York Ave, New York, NY 10065, USA. ¹⁵Program in Precision Interception and Prevention, Memorial Sloan Kettering Cancer Center, 1275 York Ave, New York, NY 10065, USA. ¹⁶Marie-Josée and Henry R. Kravis Center for Molecular Oncology, Memorial Sloan Kettering Cancer Center, 1275 York Ave, New York, NY 10065, USA. ¹⁷Weill Cornell Medical College, 407 E 61st St, New York, NY 10065, USA. ¹⁸Department of Epidemiology & Biostatistics, Memorial Sloan Kettering Cancer Center, 1275 York Ave, New York, NY 10065, USA. ✉email: papaemme@mskcc.org

