## [Peer Review File · Nature Communications]

REVIEWER COMMENTS

Reviewer #1 (Remarks to the Author): Expert in clonal hematopoiesis genomics and clinical aspects

In their manuscript entitled “Interplay between chromosomal alterations and gene mutations shapes the evolutionary trajectory of clonal hematopoiesis” Gao and colleagues explore the impact of mCAs and recurrent gene mutations in a large cohort of cancer patients. While huge datasets describing gene mutations in various contexts have been generated in the last years, mCAs have been far less investigated due to more challenging technical requirements, despite their undoubtedly high relevance for adverse clinical conditions in terms of hematologic malignancies. Therefore, the paper by Gao and colleagues provides valuable, comprehensive data not only on the frequency and characteristics of mCAs in cancer patients, but also on the interplay of mCAs with gene mutations. The authors provided detailed supplementary information, not only regarding their own data, but also putting their work into the context of previous studies on mCAs.

The manuscript is written in a concise style and figures are of high quality.

Therefore, this manuscript is a valuable and novel contribution to both researchers and clinicians in the field.

Especially with regard to translational/clinical aspects, I would presume that the data could be more exploited:

1) Clonal hematopoiesis is known to be associated with increased cardiovascular comorbidity and shorter overall survival. Is this also true for the dataset described here?

2) As shown in Supp. Tab 2, mCA+ does not affect overall blood counts. But what about those individuals with both GM and mCA+? cis versus trans? Or those that later developed hematologic disease? Were blood counts altered in these subcohorts indicative of later disease (i.e. lymphocytosis in later CLL, etc.)?

3) On the same note, I am a bit concerned that though hematologic disease was diagnosed at least 3 months after sampling, hematologic disease might have been overlooked earlier. Blood counts are only provided for five patients with MDS (Fig. 4b and Suppl. Fig. 6), but what about the rest?

4) You state that information regarding previous cancer therapies was available for a subset of patients. Does this study also include therapy-naïve cancer patients?

Integrative analysis of gene mutations and mCAs leads to interesting findings regarding cis and trans co-occurrence of the genetic aberrations.

1) Line 220: What makes you sure that the described EZH2 mutation was there before CNLOH on chr7q? Did you perform clonal evolution simulations for the described cases? Do VAFs support the notion that mutations occur in the same clone? Please provide details. As this is one of the major findings of the manuscript, single-cell data should be provided for selected cases. CNLOH have been shown to be major contributors for clonal diversification in MPN and during disease progression (Mylonas et al, Nat Commun 2020), is this also present at the CH stage?

2) With respect to mCA and GM interactions, the authors should take into account that CH can be oligoclonal with opposing evolution and differential differentiation into hematopoietic cell fractions of individual clones (see Buscarlet, Blood 2019 and Arends, Leukemia 2018). How do the authors take CH patients with multiple SNVs into account? SNV data already showed frequent bi-allelic hits (e.g. TET2-TET2).

3) Line 249: “Complex karyotype”: To my knowledge, this term is used to describe 3 or more cytogenetic aberrations but not multiple mCAs in general.

Reviewer #2 (Remarks to the Author): Expert in clonal hematopoiesis, intratumour heterogeneity, and computational genomics

In this paper Gao et al. performed deep targeted sequencing on peripheral blood samples from ~30,000 cancer patients to identify clonal haematopoiesis, identifying both mosaic chromosomal alterations (mCAs) as well as acquired gene mutations (GMs). They find evidence of mCAs acting in concert with acquired gene mutations both in cis (where the mCA affects the same locus as the gene mutation e.g. LoF GM in DNMT3A followed by a deletion on the other chromosome at overlapping position of DNMT3A) and in trans (where the mCA affects a different locus from the GM). They show that mCAs are associated with an increased risk of developing a blood cancer and argue that mCAs are an important risk factor especially when detected in combination with GMs.

The central finding, that mCAs interact with acquired mutations to shape the evolution of CH, is very exciting and builds nicely on recent work from Loh et al. showing that the interaction of mCAs and germline variants shapes the evolution of CH. There are some limitations: in most cases the data is unable to determine mutation co-occurrence in the same clone conclusively and the limited sensitivity and breadth of the mCA calling means the numbers of events with strong associations is modest. Nevertheless, I found the evidence in support of the interaction / co-occurrence hypothesis to be compelling: some specific mCAs are strongly associated with specific GMs and these often recapitulate known associations from blood malignancies. While the fact that mCAs are risk factors for blood malignancies has been shown (with more power) in Loh et al. 2020, the key finding here is that this risk increases substantially if mCAs and GMs are found together in the same individual. This finding is significant and, I believe, novel. Overall I think this is an excellent paper and should certainly be published. Naturally I have some comments listed below but these are reasonably minor. I congratulate the authors on a nice piece of work.

Sensitivity and accuracy of mCA calls

The sensitivity of the mCA calls is somewhat limited both because of a relatively small panel and because the method loses sensitivity below 10% cell fractions (where the vast majority of mCAs are - see Loh et al. 2020). I think the paper would benefit from some analysis / discussion of how much is missed due to these limitations.

Related to this, how robust are the cell fraction estimates? While the paper mentioned that replicates were performed, it did not show the correlations of inferred cell fractions. Extended Data Figure. 1 claims to show reliability of detection but this would be better shown using a scatter plot of inferred cell fractions across replicates. This way the reader can get a sense of the errors on the cell fraction measurements which are important when assessing other figures e.g. Extended Data Figure 8.

Multiple mutants clones versus multiple independent single-mutant clones?

As far as I understood, co-occurrence of mCAs and GMs in the same clone could only be conclusively

established in a relatively small number of cases where the pigeon-hole principle can be applied. In cases where this cannot be applied by where a specific mCA and GM are strongly associated it seems likely that they are in the same clone also. However, what can be said about the remaining cases? For relatively small cell fractions it seems more likely there would be 2 independent single-mutant clones - it would be nice to see more analysis discussion of these points. Are there specific mCAs that drive clonal expansions when acting alone? Are there strong associations between two or more mCAs?

Prevalence numbers

The paper would be improved if it was made clearer that the prevalence numbers quoted throughout the paper depend on the sensitivity of the assay and have little biological or clinical meaning in their own right. The same should be highlighted in reference to Extended Data Figure 7: it is not clear that the prevalence across age is actually lower for mCAs than it is for GMs: the mCA calling likely has a large false negative rate at $\leq 10\%$ (see peak in density at 0.25 cell fraction in panel a of the figure - an artefact of sensitivity limitations). A fairer comparison would be to compare both at cell fractions > 0.25 where it looks like false negatives are less of an issue.

Order of events in multiple mutants and cell-fraction / VAF correlations

I was a little confused by the statement in the main text: "cellular fractions of these cis multi-hit events generally supported homozygosity of the mutant allele and shared clonal origin" and the associated Extended Data Figure 8. If I understood it correctly, I believe the dashed lines in Extended Data Figure 8 are what would be expected if the events were either synchronous or if the double mutant completely outcompetes the single-mutant that gave rise to it. However this seems unlikely to me. Most of the time I would assume that an acquired mutation enters (forming a single-mutant clone) which then at some point later acquires a mCA (forming a double-mutant clone) with the single-mutant clone generally remaining. Therefore it is not clear why one would be constrained to a line — it should be a region. In the cases where the mutations have strong associations, deviation from the line ought to be able to be used to determine likely mutation order. I assume this is challenging because of the limited frequency resolution of the mCA data but it would have been nice to see it discussed or a few cases highlighted where you can be more confident of the order. For example in cases where you get LoF mutation in TET2 co-occurring with CNLOH, presumably the order is GM followed by mCA (since otherwise CNLOH would have no effect) - is the data consistent with this ordering? In cases involving a deletion in cis (e.g. biallelic loss of DNMT3A) or the events in trans its not clear to me what the order would be.

Reviewer #3 (Remarks to the Author): Expert in clinical research of leukaemias and myeloid neoplasms

Gao and colleagues examine the relationship between chromosomal aberrations, gene mutations and development of hematological diseases.

Technically, the manuscript is absolutely sound and reflects the great expertise the group of Papaemmanuil has. So from my point of view, all the conclusions are sound. Also the fact that chromosomal aberrations cooperate with mutations is not completely unexpected but very nice to be demonstrated

However I do not agree with the clinical implications.

1) As this is a study with clinical observation, some kind of consortial trial had been helpful, which patients had been selected how for this analysis.

2) Were the blood samples obtained before start of oncology treatment or after start of treatment or even at the end of treatment? Or are different blood samples compared at different time points? It would be very valuable to see for more patients the route of clonal evolution before and after therapy

3) Were those changes present in the blood before onset of therapy or did they appear after therapy.

This would be extremely interesting to see the impact of therapy on mutational load/ chromosomal changes in blood cells.

4) Following up on this, could it be that presence of mutational changes in peripheral blood before onset of therapy is an indication of the overall mutational burden of organism and might predict overall susceptibility to leukemia development.

5) The follow up for blood diseases in this cohort is rather very short and should be at least 5-10 years and any . The authors report onset of multiple myeloma. Multiple myeloma always arises of monoclonal gammopathy or smouldering myeloma and this precedes the onset of multiple myeloma several years. So the condition might have been present before. The same also applies for NHL. May be this part should be discussed in depth with the other coauthors who are excellent clinician scientists

6) The method for follow up for blood disease using billing information can be associated with high degree of bias (patients might not be able to afford additional treatment, might go another center) and is rather short.

7) My biggest concern regards Extended data figure 6:

It is really interesting that beam irradiation shall increase onset of mCA and cytotoxic therapy shall reduce this risk (Odds ratio smaller than 1, albeit with a high CI).

This just can not be. Irradiation therapies are really diverse but it has been shown in many studies, that chemotherapy is the one force driving leukemia development. With this results, I do not know whether there is a strong confounding bias for selection of patients and interpretation of data.

8) The fact that cancer therapy predisposes to leukemia development is not really novel. The authors report an interesting clonal evolution but what would be the really new information?

The impact of different therapies on clonal evolution would be interesting but this would be

needed to demonstrate in more detail

Or mutational burden before onset of therapy as an indication of potential overall mutational burden would be an interesting point.

The association of certain mCA and different hematological diseases would be interesting.

RESPONSE TO REVIEWERS

Reviewer #1 (Remarks to the Author): Expert in clonal hematopoiesis genomics and clinical aspects

In their manuscript entitled “Interplay between chromosomal alterations and gene mutations shapes the evolutionary trajectory of clonal hematopoiesis” Gao and colleagues explore the impact of mCAs and recurrent gene mutations in a large cohort of cancer patients. While huge datasets describing gene mutations in various contexts have been generated in the last years, mCAs have been far less investigated due to more challenging technical requirements, despite their undoubtedly high relevance for adverse clinical conditions in terms of hematologic malignancies. Therefore, the paper by Gao and colleagues provides valuable, comprehensive data not only on the frequency and characteristics of mCAs in cancer patients, but also on the interplay of mCAs with gene mutations. The authors provided detailed supplementary information, not only regarding their own data, but also putting their work into the context of previous studies on mCAs. The manuscript is written in a concise style and figures are of high quality. Therefore, this manuscript is a valuable and novel contribution to both researchers and clinicians in the field.

We thank the reviewer for the positive assessment of our study and insightful comments, which have helped us improve the manuscript.

Especially with regard to translational/clinical aspects, I would presume that the data could be more exploited:

1) Clonal hematopoiesis is known to be associated with increased cardiovascular comorbidity and shorter overall survival. Is this also true for the dataset described here?

The reviewer is right that CH (in the form of gene mutations) has also been associated with atherosclerotic disease (Jaiswal et al, NEJM 2017), and has also been shown to correlate with lower overall survival in prior cancer studies (Coombs et al, Cell Stem Cell 2017; Bolton et al, in press, doi:<https://doi.org/10.1101/848739>).

With regards to cardiovascular comorbidity, a recent large-scale population based study investigating the potential impact of mCA has not identified significant associations (Loh et al, Nature 2020). Given the larger cohort size (n=482,789), their study was better powered to detect any potential association between mCA and cardiovascular disease compared to our study. Additionally, clinical annotation for subsequent cardiovascular diseases in our cohort was not readily available or would have required substantial data curation. Thus, we decided that the evaluation of cardiovascular comorbidities was out of the scope of this study, whose main focus is

the evolutionary trajectories of clonal hematopoiesis, their role in leukemic transformation and how these insights inform upon the clinical management of cancer patients.

With regards to overall survival, we indeed observe a significant association of gene mutations with shorter overall survival ($p < 0.0001$) but not for mCAs ($p = 0.064$) (please see figures included below). However, our interpretation is that CH is a biomarker of many adverse risk factors for OS (i.e. age, tumor type, disease stage, smoking, prior treatment, treatment intensity) that are potential confounders. Due to these limitations in generating robust conclusions from this data, we decided to not include data on OS in this manuscript.

Overall survival in patients with and without CH gene mutations and mCAs. (a) Overall survival in patients with and without gene mutations. **(b)** Overall survival in patients with and without mCAs.

2) As shown in Supp. Tab 2, mCA+ does not affect overall blood counts. But what about those individuals with both GM and mCA+? cis versus trans? Or those that later developed hematologic disease? Were blood counts altered in these sub-cohorts indicative of later disease (i.e. lymphocytosis in later CLL, etc.)?

The reviewer raises two important questions: first, whether mCA and its co-mutations affect blood count compositions and second, whether blood count abnormalities in certain patient subgroups can be indicative of incipient or active hematological disease present at CH assessment in our cohort.

We agree that the effect of CH mutations on blood counts is an interesting point to investigate. This has been nicely demonstrated for mCAs in prior studies (Loh et al, Nature 2018 and Loh et al, Nature 2018). However, it has also been shown that certain CH mutations (e.g. those in *TP53*, *PPM1D*, *CHEK2*) are promoted by oncologic therapy (Bolton et al, in press, doi:<https://doi.org/10.1101/848739>). Since oncologic therapy also strongly impacts blood counts in cancer patients, we could not be sure whether blood abnormalities that we see in our cohort are

a consequence of mCA/GM co-mutations or a result of confounding by recent therapy. Therefore our dataset is not well-suited to address this question.

On the other hand, it is indeed possible that blood counts can be indicative of incipient or active disease unknown at the time of blood collection. However, as discussed above blood count abnormalities can result from a variety of causes, including but not limited to recent oncologic therapy, nutritional deficiencies, chronic disease, or genetic predisposition. Even in the presence of detectable mutations, a definitive diagnosis cannot be made without fulfilling the established morphological criteria (with the exception of a few disease-defining alterations) for diseases such as MDS (Steensma et al, Blood 2015; Cargo et al, Blood 2015). Therefore, it is unclear whether blood abnormalities at the time of CH assessment in these sub-cohorts truly reflect incipient or active disease without ruling out other potential causes and going through a retrospective diagnostic workup. For the purpose of this study, we adopted the assessment of the oncologists at the time. We will more comprehensively address the reviewer's question about the blood counts of different sub-cohorts and present the requested data in response to the next question.

3) On the same note, I am a bit concerned that though hematologic disease was diagnosed at least 3 months after sampling, hematologic disease might have been overlooked earlier. Blood counts are only provided for five patients with MDS (Fig. 4b and Suppl. Fig. 6), but what about the rest?

We would like to acknowledge the reviewer's point and highlight that accurate and timely diagnosis of hematologic disease in cancer patients undergoing therapy represents a significant clinical challenge. It is indeed possible that a small number of patients with hematologic diseases that were unknown at the time of sequencing are present in the cohort. We will first discuss our procedure for capturing hematologic disease diagnoses and then evaluate the potential impact of the presence of these patients on our conclusions.

We tried to be as comprehensive as possible in excluding patients with a known hematologic disease. Our procedure includes:

1. Excluding subjects who had a billing code (endorsed by any provider, including outside institutions) related to hematologic malignancy prior to, or less than 3 months after the time of blood draw.
2. Excluding subjects with any record of hematologic disease in the tumor registry, based on a review of an entire patient chart once per 6 months after initial diagnosis.
3. Reviewing the electronic medical records (including physician visit notes, hematopathology reports, and molecular testing results) of mCA positive patients to ensure that these patients did not have a prior hematological malignancy before/during receiving care at MSK.
4. Reviewing electronic medical records of patients who received NGS-based assay for hematologic malignancies (MSK-hemepact) and excluding any patient with a positive diagnosis.

As the reviewer correctly pointed out, there still might be a small number of patients with active or incipient heme disease that either were not evaluated by the current standards of surveillance or underwent workup that did not result in a definitive diagnosis. For the purpose of this study, we decided to adopt the assessment of the oncologists at the time. We believe that the insights from this study provides the foundation for establishing NGS-based CH screening as a more sensitive biomarker for incipient hematologic disease.

In the first part of our study, we describe the co-occurrence of mCAs and gene mutations in CH. To ensure that the co-mutations that we report are not entirely accounted for by patients with incipient disease, we analyzed the blood counts of the subset of subjects who harbored one of the recurrent co-mutations at the time of CH assessment (added as Supplementary Fig. 7). This showed that most of the subjects with recurrent CH co-mutation (57 out of 70) have normal or near-normal range blood counts at the time (we defined normal or near-normal range blood count as not fulfilling the current WHO criteria for cytopenia and absolute lymphocyte count $> 5 \times 10^9/L$), and that each type of co-mutations that we highlight in the results are observed in at least one subject with normal or near-normal blood counts. We have added the following sentence in the results section “Patterns of co-occurrence between mCAs and gene mutations reveal diverse mechanisms of selection” to reference this analysis.

“Interestingly, most of the subjects with recurrent CH co-mutations do not exhibit significant blood abnormalities at the time of blood draw (Supplementary Fig. 7)”

Comutation	Cytopenia	Lymphocytosis	Normal	Total
(chr11CNLOH chr11DEL) & ATM	1	0	2	3
(chr4DEL chr4CNLOH) & TET2	1	0	8	9
chr12AMP & (NOTCH1 MYD88 FBXW7)	0	1	5	6
chr13DEL & ATM	1	0	1	2
(chr17CNLOH chr17DEL chr17AMP) & TP53	1	2	1	4
chr1CNLOH & MPL	2	0	2	4
chr2ODEL & (U2AF1 TERT)	1	0	4	5
chr2DEL & DNMT3A	0	0	5	5
chr3AMP & TP53	0	0	2	2
chr4DEL & (SRSF2 ATM CHEK2 ASXL1)	0	0	4	4
chr5DEL & TP53	2	0	3	5
chr7DEL & (TP53 PPM1D)	0	0	1	1
chr7CNLOH & EZH2	1	0	5	6
chr8AMP & TET2	0	0	3	3
chr9CNLOH & JAK2	0	0	11	11
Total	10	3	57	70

Supplementary Fig. 7: Blood count abnormalities in individuals with recurrent co-mutations. Statuses of cytopenia and lymphocytosis were determined using the lab test closest to the timepoint of CH assessment according to the WHO criteria (anemia: hemoglobin $< 10g/dL$, thrombocytopenia: platelets $< 100 \times 10^9/L$, neutropenia: absolute neutrophil count $< 1.8 \times 10^9/L$, lymphocytosis: absolute lymphocyte count $> 5 \times 10^9/L$).

Under co-mutation category, “|” indicates OR relationship, “&” indicates AND relationship between the mutations.

CH status	Cytopenia	Lymphocytosis	Normal	Total
GM	1315	14	8308	9637
mCA	13	0	116	129
mCA + GM	33	7	177	217
No CH	2963	10	19486	22459

Blood count abnormalities in individuals with mCAs and gene mutations. GM, gene mutation. Statuses of cytopenia and lymphocytosis were determined according to the criteria described above.

In the second part of the study, we demonstrate that mCA is a significant independent risk factor for leukemia transformation through a multivariate cause-specific Cox regression analysis. To ensure that this result is not driven by patients with incipient leukemia at the time of CH assessment, we performed landmark analysis (Extended Data Fig. 10) and showed that even among patients who were free of hematological disease within one year of assessment, mCA remains a significant independent predictor of leukemia diagnosis. In addition, we compiled the prevalence of cytopenia and lymphocytosis in the subset of patients that were later diagnosed with hematological neoplasms, as shown in the table below.

Disease	Cytopenia	Lymphocytosis	Normal	Total
CLL	1	3	8	12
Lymphoid	2	0	27	29
Myeloid	6	0	42	48
None	3674	24	24703	28401

Blood count abnormalities in individuals with subsequent hematological neoplasm diagnosis. Statuses of cytopenia and lymphocytosis were determined according to the criteria described above.

This showed that the patients who were diagnosed with myeloid or lymphoid neoplasms did not exhibit elevated rates of blood abnormalities at the time of CH assessment, while those diagnosed with CLL showed an enrichment for lymphocytosis. Given the chronic nature of CLLs, it is possible that the disease of these patients were overlooked at the time. To address the reviewer’s concern, we reperformed the multivariate cause-specific Cox regression analysis (Figure 4d) while restricting to patients with normal blood counts (no cytopenia or lymphocytosis) at the time of CH assessment. We obtained similar hazard ratio estimates for gene mutations as well as mCAs. We have accordingly added a sentence in the results section titled “Implications for risk stratification” to reference this analysis:

“In addition, we obtain similar risk estimates for gene mutations and mCA after restricting the analysis to patients with no blood abnormalities at the time of CH assessment (Supplementary Fig. 11).”

Supplementary Fig. 11: Hazard ratios of CH mutations in individuals with and without blood abnormalities. Hazard ratios and 95% CIs of CH features in a multivariable cause-specific Cox regression model with a 9 month landmark. The HR for 10% increment in putative driver (PD) VAF was shown. mCA status was included as a binary indicator.

In summary, we acknowledge that while it is indeed possible that individuals with unknown hematological disease are present in the cohort, their presence would not affect our main conclusions in the study. Furthermore, it is of clinical interest to include those patients with occult or incipient diseases that are hard to identify with the current clinical care. We have added the following sentence to the discussion section to emphasize this point.

“Beyond individuals at risk, our data supports that there is an opportunity to also identify individuals with suspect hematologic disease that are difficult to capture with current clinical surveillance.”

4) You state that information regarding previous cancer therapies was available for a subset of patients. Does this study also include therapy-naïve cancer patients?

Our study cohort indeed contains therapy-naïve cancer patients. As MSKCC is frequently a referral center, treatment histories for patients who have previously received clinical care at external centers are difficult to obtain. Thus, to study the effect of therapy on mCA risk we restricted our analysis in patients who received all their clinical care at MSKCC (therapy cohort). We extracted complete treatment information for 10,375 (out of 32,442) patients among whom 4,125 were free of prior receipt of external beam radiation (XRT) or cytotoxic chemotherapy while 6,250 received either XRT or cytotoxic chemotherapy. The remaining 22,067 patients represent a mixture of previously treated and therapy-naïve patients, however this information is not readily available.

Integrative analysis of gene mutations and mCAs leads to interesting findings regarding cis and trans co-occurrence of the genetic aberrations.

1) Line 220: What makes you sure that the described *EZH2* mutation was there before CNLOH on chr7q? Did you perform clonal evolution simulations for the described cases? Do VAFs support the notion that mutations occur in the same clone? Please provide details. As this is one of the major findings of the manuscript, single-cell data should be provided for selected cases. CNLOH have been shown to be major contributors for clonal diversification in MPN and during disease progression (Mylonas et al, Nat Commun 2020), is this also present at the CH stage?

We thank the reviewer for this query. We reply on three direct lines of evidence in our data to infer that *EZH2* mutations likely arose before 7qCNLOH and that they likely reside in the same clone: high variant allele frequencies indicating homozygosity, paired leukemia samples showcasing concurrent clonal expansion, and the cohort-wide distribution of *EZH2* mutations and 7qCNLOH events. First, using a combination of CH samples and paired leukemia samples at a later time point, we could confidently infer the homozygosity of the *EZH2* mutation through its high variant allele frequency in 3 out of 6 cases of *EZH2* double hits. Specifically, patient P-0006225 had *EZH2* VAF at 48% in the CH sample and 93% in the MDS sample (binomial test, null hypothesis VAF = 50%, $p < 2.2e-16$). Patient P-0002903 had *EZH2* VAF at 64% in the CH sample and 88% in the MDS sample (binomial test, null hypothesis VAF = 50%, $p < 2.2e-16$). Patient P-0004831 had *EZH2* VAF of 75% (binomial test, null hypothesis VAF = 50%, $p < 9.3e-11$) in the CH sample. For the remaining 3 cases of *EZH2*/7qCNLOH in the low VAF range and for which we do not have a paired sample, we can observe that their VAF and mCA cell fraction estimates are at a ratio that is consistent with loss of heterozygosity (green diagonal line versus gray line of Extended Data Fig. 8). Based on the infinite sites assumption, loss of heterozygosity generally necessitates the mutation order of *EZH2* followed by CNLOH. Second, in patient P-0002903 and P-0006225 (Fig. 4b), we observe concurrent clonal expansions of *EZH2* mutations and 7qCNLOH events, both achieving clonal status in the subsequent leukemia sample. This implies that both mutations were present in the same cell population also at the CH stage. Third, on a population scale, we observe *EZH2* CH mutations in the absence of 7qCNLOH ($n=37$) but not 7qCNLOH in the absence of a *EZH2* mutation ($n=0$). This suggests that *EZH2* mutations likely precede the acquisition of 7qCNLOH in the evolution of CH. We would also like to add that our observation is consistent with prior evidence that in diagnostic samples at MPN, AML and MDS, *EZH2* frequently achieves homozygosity through 7qLOH (Ogawa MDS, Papaemmanuil Blood 2013, Papaemmanuil, NEJM 2015, Nangalia, NEJM 2013, Yoshizato Blood 2013). We agree with the reviewer that the basis for this statement should be made more transparent in the text, and have thus rephrased the sentence as following.

“The high variant allele fractions and the absence of 7qCNLOH events without a concurrent *EZH2* mutation suggest that 7qCNLOH is a highly directed event targeting a previously acquired mutation of *EZH2* in CH (Fig. 3b, Extended Data Fig. 8a).”

The reviewer also raises the interesting question whether concurrent acquisitions of CNLOH can lead to clonal diversifications in CH. However, due to the difficulty in inferring multiple subclonal alterations at low cell fractions at the same genomic site in targeted sequencing data, as well as the

unavailability of viable cells to conduct single-cell experiments, this question would be out of scope of our current study and would be a topic for future research.

2) With respect to mCA and GM interactions, the authors should take into account that CH can be oligoclonal with opposing evolution and differential differentiation into hematopoietic cell fractions of individual clones (see Buscarlet, Blood 2019 and Arends, Leukemia 2018). How do the authors take CH patients with multiple SNVs into account? SNV data already showed frequent bi-allelic hits (e.g. TET2-TET2).

The reviewer brings up three important points, which we will address separately.

First, it is indeed possible that certain cases of concurrent mCA and GM identified in this study may have arisen in separate and competing HSC cell populations. However the data in this regard are sparse, and larger studies are warranted to inform on the prevalence and relevance of oligoclonality in CH, which we trust that the reviewer will agree are beyond the scope of this study. Importantly most of the GM-mCA interactions we describe here in CH are well studied in leukemia genomes (Grinfeld, NEJM 2018; Papaemmanuil, Blood 2013; Ogawa, Blood, 2019; Graubert, Nature Genetics 2011; Papaemmanuil, NEJM 2016). These studies show that the same recurrent co-mutation events occur in the same cells as manifested by the concomitant dominant cell fractions at leukemia diagnosis. The global patterns of highly specific and recurrent co-mutation patterns presented in the current study allowed us to infer that these previously established mechanisms of clonal selection are also operative in CH. Therefore, even if present, the possibility of oligoclonal CH would not impact the conclusions in this study in a major way.

Second, as Arends and colleagues have also shown, there may be CH cases that display differential representation of mutations amongst distinct hematopoietic cells. Since our study focuses on clonal proportions of CH mutations in whole bloods, this would be out of scope of the present study. Differential representation of these events in other parts of the hematopoietic system is unlikely to influence the main results of this study and importantly would be difficult to resolve in the absence of viable frozen sorted cell populations, which we do not have access to. Acknowledging that there are a number of scenarios that bulk sequencing cannot fully resolve as compared to single cell studies, we refer the reviewer to this sentence in the discussion:

“Future development of technology platforms that can reliably map gene mutations and chromosomal aberrations jointly at a single cell resolution are warranted to fully resolve their clonal relationships.”

Third, the reviewer raises the issue of concomitant gene mutations, which have also been reported in a subset of individuals with CH. In our cohort 2935 (9% of the cohort) subjects had 2 or more gene mutations, whereas 787 had multiple SNVs/indels in the same gene. *DNMT3A*, *TET2*, and *PPM1D* exhibited notable patterns of multiple concomitant gene mutations (please see figure below). Although it is unclear whether these mutations reside in the same cells (Arends, Leukemia 2018), the frequent occurrence of concomitant mutations affecting the same gene further hints at

the selective advantage of double mutants, and/or convergent evolution under selection pressure. This is an interesting point that we have investigated in two separate studies (Bolton et al 2019, in press, Miles et al 2020, in press). Since the main focus of this study has been to study the interplay of mCAs and gene mutations, we reviewed multiple gene mutations in the context of mCAs that appeared in *cis*. Out of 45 CH cases that exhibited double hits in the form of mCA and gene mutation in *cis*, 5 had two distinct SNVs/indels in the same gene (2 cases in *DNMT3A*, 1 case in *TET2*, 2 cases in *EZH2*). Given that this is a minor proportion of the *cis* double hit events described in the study, there is not substantial evidence that suggests concomitant gene mutations will impact the conclusions of the study, which highlight known mechanisms of genetic interactions between mCA and gene mutations.

To make the presence of concomitant gene mutations more clear when represented visually, we have highlighted cases of multiple gene mutations in close proximity in Fig. 3b, and have clarified this change in the figure description.

“Gene loci that exhibited recurrent double hits in *cis*. Zoom window shows the gene loci affected and the location of the gene mutations. “2” denotes two adjacent mutations in close proximity with each other.”

Occurrences of concomitant CH mutations in the same gene. Stacked bars are colored by the number of mutations in the same gene.

3) Line 249: “Complex karyotype”: To my knowledge, this term is used to describe 3 or more cytogenetic aberrations but not multiple mCAs in general.

The reviewer is correct. We revised this sentence to the following.

“In addition, we observe an significant association of *TP53* mutations with **the presence of multiple concomitant mCAs** in CH patients (Fisher’s exact test, $p = 0.0015$), as evidenced by the presence of simultaneous alterations across multiple chromosomes (Extended Data Fig. 9).”

Reviewer #2 (Remarks to the Author): Expert in clonal hematopoiesis, intratumour heterogeneity, and computational genomics

In this paper Gao et al. performed deep targeted sequencing on peripheral blood samples from ~30,000 cancer patients to identify clonal haematopoiesis, identifying both mosaic chromosomal alterations (mCAs) as well as acquired gene mutations (GMs). They find evidence of mCAs acting in concert with acquired gene mutations both in cis (where the mCA affects the same locus as the gene mutation e.g. LoF GM in DNMT3A followed by a deletion on the other chromosome at overlapping position of DNMT3A) and in trans (where the mCA affects a different locus from the GM). They show that mCAs are associated with an increased risk of developing a blood cancer and argue that mCAs are an important risk factor especially when detected in combination with GMs.

The central finding, that mCAs interact with acquired mutations to shape the evolution of CH, is very exciting and builds nicely on recent work from Loh et al. showing that the interaction of mCAs and germline variants shapes the evolution of CH. There are some limitations: in most cases the data is unable to determine mutation co-occurrence in the same clone conclusively and the limited sensitivity and breadth of the mCA calling means the numbers of events with strong associations is modest. Nevertheless, I found the evidence in support of the interaction / co-occurrence hypothesis to be compelling: some specific mCAs are strongly associated with specific GMs and these often recapitulate known associations from blood malignancies. While the fact that mCAs are risk factors for blood malignancies has been shown (with more power) in Loh et al. 2020, the key finding here is that this risk increases substantially if mCAs and GMs are found together in the same individual. This finding is significant and, I believe, novel. Overall I think this is an excellent paper and should certainly be published. Naturally I have some comments listed below but these are reasonably minor. I congratulate the authors on a nice piece of work.

We thank the reviewer for recognizing the novelty of our findings as well as relevance to the field.

Sensitivity and accuracy of mCA calls

The sensitivity of the mCA calls is somewhat limited both because of a relatively small panel and because the method loses sensitivity below 10% cell fractions (where the vast majority of mCAs are - see Loh et al. 2020). I think the paper would benefit from some analysis / discussion of how much is missed due to these limitations.

We thank the reviewer for bringing up this great point. Upon re-review, we realize there is indeed an opportunity to draw insights into types of events that would not be captured by our detection method using data from prior studies. We added this analysis that compares the characteristics of events with cell fraction below 10% and events with cell fraction above 10% in Loh et al, Nature 2018 as Supplementary Fig. 3. Indeed, there are more mCA events present below 10% cell fraction. However, the alteration types and genomic distributions of events are highly consistent across cell fractions, with the exception of a few events that are preferentially present in low cell fractions (e.g. CNLOH on chr2, chr3, chr8), which would not be captured in our study. We refer to this analysis in the results section titled “Landscape of mCAs in prospective sequencing of cancer patients”:

“The genomic distribution and patterns of the mCAs we detected using a targeted sequencing platform with sparse genome-wide coverage were highly consistent with findings from high-density SNP array studies, confirming the specificity of our approach^{13,14,16} (Fig. 1e, Extended Data Fig. 3-5, **Supplementary Fig. 3**).”

We have additionally modified our discussion to reflect this limitation:

“Since our methodology is biased towards the detection of mCAs in relatively large clones ($\geq 10\%$), **it is likely that mCAs that are preferentially present in low cell fractions** or in regions not optimally tiled by our panel may have been missed (**Extended Data Fig. 3, Supplementary Fig. 3**).”

Supplementary Fig. 3: Genome-wide distributions of detected mCA events with different cell fractions. We compare the genomic distributions of mCAs detected at <10% cell fraction and $\geq 10\%$ cell fraction in Loh et al. 2018 and in this study. Events are colored by alteration type (AMP, amplification; CNLOH, copy-neutral loss of heterozygosity; DEL, deletion).

Related to this, how robust are the cell fraction estimates? While the paper mentioned that replicates were performed, it did not show the correlations of inferred cell fractions. Extended Data Figure. 1 claims to show reliability of detection but this would be better shown using a scatter plot of inferred cell fractions across replicates. This way the reader can get a sense of the errors on the cell fraction measurements which are important when assessing other figures e.g. Extended Data Figure 8.

We thank the reviewer for making this suggestion. We performed the proposed analysis and observed that our cell fractions estimates are highly concordant across replicates with some degree of variability. We have added this result as Supplementary Fig. 2 and provided a reference in the results section “Landscape of mCAs in prospective sequencing of cancer patients”.

“The precision of this approach was validated in 919 technical replicates (Methods, Extended Data Fig. 1, Supplementary Fig. 2).”

Supplementary Fig. 2: Estimated cell fractions of mCAs detected in replicate samples. Cell fraction 1, estimated cell fraction in study samples. Cell fraction 2, estimated cell fraction in replicate samples. Text labels indicate chromosome number. Colors indicate alteration type (AMP, amplification; CNLOH, copy-neutral loss of heterozygosity; DEL, deletion). Diagonal line indicates equal estimated cell fractions in both samples.

Multiple mutants clones versus multiple independent single-mutant clones?

As far as I understood, co-occurrence of mCAs and GMs in the same clone could only be conclusively established in a relatively small number of cases where the pigeon-hole principle can be applied. In cases where this cannot be applied by where a specific mCA and GM are strongly associated it seems likely that they are in the same clone also. However, what can be said about the remaining cases? For relatively small cell fractions it seems more likely there would be 2 independent single-mutant clones - it would be nice to see more analysis discussion of these points. Are there specific mCAs that drive clonal expansions when acting alone? Are there strong associations between two or more mCAs?

The reviewer is correct that in this study, the co-occurrence of mCAs and GMs in the same clone can be established in only a subset of cases (13 out of 44) using the pigeonhole principle (Extended Data Fig. 8). Based on prior knowledge from studies of leukemia genomes (Grinfeld, NEJM 2018; Papaemmanuil, Blood 2013; Ogawa, Blood, 2019; Graubert, Nature Genetics 2011; Papaemmanuil, NEJM 2016) we were able to infer that the recurrent co-mutations that we observe in CH resembling known interactions are also likely to operate in the same clones. For the remaining cases (co-mutations that are neither recurrent or amenable to pigeonhole principle), we thought it would be hard to draw conclusions on the underlying clonality. It is indeed possible for them to

arise in separate clones (Arends et al, Leukemia 2018). Therefore, we decided to not highlight these cases in the results. We cover the possibility of polyclonality in the discussion:

“Future development of technology platforms that can reliably map gene mutations and chromosomal aberrations jointly at a single cell resolution are warranted to fully resolve their clonal relationships.”

The reviewer raises a good point that certain mCAs indeed tend to drive clonal expansions alone. As one can observe from Fig. 3a, deletions in chrX (23 out of 48) and chr20 (21 out of 48) frequently occurred in isolation without any clear co-mutations. The same can be said for gains in chr15 (5 out of 10) and chr12 (4 out of 16), as well as CNLOH in chr11 (6 out of 12), chr14 (6 out of 9), and chr22 (6 out of 9), although these events are less frequently observed. We have added the following sentence to our results section “Patterns of co-occurrence between mCAs and gene mutations reveal diverse mechanisms of selection” to reflect this observation:

“It is also of note that certain mCAs (e.g. chr20-, X-) frequently occurred in isolation without any clear co-mutations, indicating that these events can also drive clonal expansions alone (Fig. 3a).”

With regards to co-occurrence between mCAs, our sample size is too small to investigate the co-occurrences of specific alterations. The only recurrent mCA pair that we were able to observe is chr12 and chr19 trisomies, which has been reported in Loh et al, Nature 2018.

Prevalence numbers

The paper would be improved if it was made clearer that the prevalence numbers quoted throughout the paper depend on the sensitivity of the assay and have little biological or clinical meaning in their own right. The same should be highlighted in reference to Extended Data Figure 7: it is not clear that the prevalence across age is actually lower for mCAs than it is for GMs: the mCA calling likely has a large false negative rate at $\leq 10\%$ (see peak in density at 0.25 cell fraction in panel a of the figure - an artefact of sensitivity limitations). A fairer comparison would be to compare both at cell fractions >0.25 where it looks like false negatives are less of an issue.

We thank the reviewer for raising this point. It is true that our estimates of the prevalence of CH mutations are influenced by a variety of factors such as detection sensitivity, genomic coverage, as well as cohort characteristics. We have added the following sentence in the results section “Global characteristics of mCAs in relation to gene mutations” to clarify this point:

“The estimates of mutation prevalence are dependent on the detection sensitivity and genomic coverage specific to the sequencing assay used in this study as well as characteristics of the MSK-IMPACT patient cohort.”

With regards to the population incidence of CH mutations presented in Extended Data Fig. 7, we agree with the reviewer that this comparison would be made fairer if we restrict to events with

inferred cell fraction > 0.25. We have added this analysis as Extended Data Fig. 7c, which compares the prevalence of mCAs and GMs at an inferred cell fraction of 25% for both types of mutations. The result still indicates that mCAs have lower prevalence across age groups than GMs. We have additionally added a sentence in the figure caption to guide the interpretation of these prevalence estimates:

“Estimates of mutation prevalence are dependent on the detection sensitivity and genomic coverage specific to our sequencing panel (MSK-IMPACT) as well as cohort characteristics such as gender, race, and mutagen exposures.”

Extended Data Figure 7c: Comparison of age-related incidence of gene mutations and chromosomal alterations with inferred cell fraction >25%. Estimates of mutation prevalence are dependent on the detection sensitivity and genomic coverage specific to our sequencing panel (MSK-IMPACT) as well as cohort characteristics such as gender, race, and mutagen exposures.

Order of events in multiple mutants and cell-fraction / VAF correlations

I was a little confused by the statement in the main text: “cellular fractions of these cis multi-hit events generally supported homozygosity of the mutant allele and shared clonal origin” and the associated Extended Data Figure 8. If I understood it correctly, I believe the dashed lines in Extended Data Figure 8 are what would be expected if the events were either synchronous or if the double mutant completely outcompetes the single-mutant that gave rise to it. However this seems unlikely to me. Most of the time I would assume that an acquired mutation enters (forming a single-mutant clone) which then at some point later acquires a mCA (forming a double-mutant clone) with the single-mutant clone generally remaining. Therefore it is not clear why one would be constrained to a line — it should be a region. In the cases where the mutations have strong associations, deviation from the line ought to be able to be used to determine likely mutation order. I assume this is challenging because of the limited frequency resolution of the mCA data but it would have been nice to see it discussed or a few cases highlighted where you can be more confident of the order. For example in cases where you get LoF mutation in TET2 co-occurring with CNLOH, presumably the order is GM followed by mCA (since otherwise CNLOH would have no

effect) - is the data consistent with this ordering? In cases involving a deletion in cis (e.g. biallelic loss of DNMT3A) or the events in trans its not clear to me what the order would be.

The reviewer is correct that the dashed lines in Extended Data Fig. 8 represent the expected ratios of gene mutation VAF and mCA cell fractions if there only exists one clonal cell population that contains both mutations, and that cases with a mixture of single and double mutants would not be constrained by this line. We thank the reviewer for clarifying this point. We think that the dashed lines serve as useful visual references despite that they do not cover all possible scenarios, and therefore have modified the figure caption to further guide interpretation:

“Dashed lines indicate the expected ratios of mCA cell fraction and gene mutation VAF if they co-occur in the same clone in cis (red: deletion, green: CNLOH) or in trans (gray), **provided that this clone constitutes the whole mutant cell population and that the wild type allele is deleted in cases of cis double hits.**”

We agree with the reviewer that the mutation acquisition order is worth further investigation, and that our results are indeed consistent with the proposed sequence of events. In 10 double hit cases involving a CNLOH (highlighted as solid dots in Extended Data Fig. 8), we can infer that the CNLOH and the gene mutation in *cis* likely reside in the same clone using the pigeonhole principle ($CF_{mCA} + VAF > 1$). Out of these cases, all 10 have a VAF significantly higher than 0.5 (two-tailed binomial test, $p < 0.01$), indicating homozygosity of the variant and the likely mutation order of GM followed by CNLOH (please see table below).

The reviewer raises another point about determining the likely mutation order using deviation from the expected cell fraction ratio. Out the 10 double hit cases discussed above, 4 (2 *TET2*/4qCNLOH, 1 *EZH2*/7qCNLOH, and 1 *JAK2*/9pCNLOH) showed markedly lower CNLOH cell fractions (two-tailed binomial test, $p < 0.01$) than the corresponding gene mutation VAF. This indicates ongoing clonal evolution where a double mutant (through CNLOH) clone co-exists with an ancestral clone of single mutant cells, further supporting the mutation order of GM followed by CNLOH (please see table below). While this observation is indeed interesting, we would like to note that the uncertainty in the inferred mCA cell fraction estimation is unknown, so a rigorous statistic for the deviation from the expected cell fractions would be hard to derive (like the reviewer has pointed out). We therefore think it is prudent to not draw conclusions from these few cases and leave to future studies to more conclusively determine the mutation acquisition order. However, we have added a sentence to the caption of Extended Data Fig. 8 to aid interpretation:

“Solid dots represent pairs of mCAs and gene mutations that can be inferred to reside in the same clone by the pigeonhole principle (sum of inferred cell fraction $> 100\%$). **In these cases, deviations from the expected mutant fractions may indicate a mixture of clones with one and both mutations, and/or noise in measurement.**”

patient_id	start	end	Gene	VariantClass	VAF_N	cellfrac_mCA	cellfrac_GM	pigeonhole	N_AltCount	N_TotalDepth	binom_0.5_p	binom_diff_p
P-0000681	422008	22006027	JAK2	snv	0.59385	0.67	0.59385	TRUE	193	325	0.0008472	0.0045504
P-0000741	66467400	190262500	TET2	snv	0.77922	0.44	0.77922	TRUE	120	154	1.90E-12	1.24E-17
P-0002500	84383901	190262486	TET2	snv	0.70417	0.73	0.70417	TRUE	169	240	2.16E-10	0.3829271
P-0002903	55270257	158450900	EZH2	snv	0.64	0.4	0.64	TRUE	343	536	9.33E-11	6.39E-29
P-0029604	421860	37020800	JAK2	snv	0.64164	0.59	0.64164	TRUE	188	293	1.43E-06	0.0748713
P-0041882	422052	27190655	JAK2	snv	0.64407	0.38	0.64407	TRUE	76	118	0.0022427	9.72E-09
P-0045590	84383761	190262408	TET2	snv	0.76147	0.74	0.76147	TRUE	249	327	5.74E-22	0.4124985
P-0047673	422000	75047061	JAK2	snv	0.61688	0.57	0.61688	TRUE	190	308	4.85E-05	0.1069593
P-0015946	75662515	190262500	TET2	indel	0.59296	0.41	0.59296	TRUE	236	398	0.0002437	2.51E-13
P-0004831	99927916	148512600	EZH2	indel	0.76562	0.74	0.76562	TRUE	294	384	2.24E-26	0.2691813

Co-occurring mCAs and gene mutations in *cis*. Only cases where pigeonhole principle can be applied are shown. Column `binom_0.5_p` indicates the significance of the GM VAF deviation from 50% obtained from a two-tailed binomial test. Column `binom_diff_p` indicates the significance of the GM VAF deviation from the co-occurring mCA cell fraction obtained from a two-tailed binomial test.

Reviewer #3 (Remarks to the Author): Expert in clinical research of leukaemias and myeloid neoplasms

Gao and colleagues examine the relationship between chromosomal aberrations, gene mutations and development of hematological diseases.

Technically, the manuscript is absolutely sound and reflects the great expertise the group of Papaemmanuil has. So from my point of view, all the conclusions are sound. Also the fact that chromosomal aberrations cooperate with mutations is not completely unexpected but very nice to be demonstrated.

However I do not agree with the clinical implications.

We are very grateful to read the positive assessment of our study methods, results and conclusions and appreciate the opportunity to clarify the reviewer's concerns.

1) As this is a study with clinical observation, some kind of consortial trial had been helpful, which patients had been selected how for this analysis.

We thank the reviewer for this suggestion. We have added a CONSORT flow diagram as Supplementary Fig. 1 that explains how patients are selected for this analysis. We reference this figure in the results section titled "Patient cohort and samples":

"The study population included 32,442 solid tumor patients (median age: 61, range 20-99, Supplementary Table 2) with non-hematologic cancers that underwent matched tumor and blood sequencing at MSKCC using the MSK-IMPACT panel²⁸ on an institutional prospective sequencing protocol (ClinicalTrials.gov number, NCT01775072) before Feb 1st 2020 (Supplementary Fig. 1)."

We have also added additional clarifications in the methods section titled "MSK-IMPACT cohort".

Supplementary Fig. 1: CONSORT diagram for cohort selection. EMR, electronic medical records. QC, quality control. For more details please refer to the methods section.

2) Were the blood samples obtained before start of oncology treatment or after start of treatment or even at the end of treatment? Or are different blood samples compared at different time points? It would be very valuable to see for more patients the route of clonal evolution before and after therapy

We thank the reviewer for this query and apologize for the lack of clarity in the original manuscript. We profiled the landscape of CH mutations from peripheral blood samples sequenced by a prospective clinical sequencing panel (MSK-IMPACT) that is typically administered at the patient's first presentation at MSKCC. The majority of the patients only received IMPACT testing once, therefore our dataset used for CH assessment only contains one sample at a single time point per patient. In terms of treatment status at the time of sequencing, our cohort includes primary diagnostic and previously untreated patients, previously treated patients, or patients undergoing therapy. Since treatment histories for patients who have previously received clinical care at

external centers are difficult to obtain, we restricted our analysis on the effect of therapy on mCA risk in patients who received all their clinical care at MSKCC (therapy cohort). This includes 10,375 (out of 32,442) patients with complete treatment information prior to blood collection, among whom 4,125 were free of prior receipt of external beam radiation (XRT) or cytotoxic chemotherapy while 6,250 have previously received either XRT or cytotoxic chemotherapy.

The reviewer raises an interesting point with regards to the effect of treatment on the route of clonal evolution, which is best answered by prospective collection of longitudinal samples in patients with mCA before, during and after treatment. However, as discussed above our analysis relies on retrospective data that mostly contains single time point samples from patients who received IMPACT testing at MSKCC from Jan 2014 to Feb 2020, and given the rarity of detectable mCAs (1% of the patients in the cohort carry a detectable mCA) it has not been possible to collect a sufficient number of sequential samples. Future studies are warranted to investigate the role of therapy in mCA selection during clonal evolution.

3) Were those changes present in the blood before onset of therapy or did they appear after therapy. This would be extremely interesting to see the impact of therapy on mutational load/ chromosomal changes in blood cells.

We agree with the reviewer that the effect of therapy on chromosomal alterations in CH and whether these mutations are pre-existing are indeed interesting points of investigation. In the subset of patients (n=10,375) with complete prior treatment histories, we detected mCA in 63 out of 6250 (1%, CI = 0.8%-1.216%) treated patients (external beam radiation or cytotoxic therapy) and in 50 out of 4125 (1.2%, CI = 0.95%-1.5%) previously untreated patients. However, given the rarity of detectable mCAs, our sample size is too small to provide conclusive evidence whether therapy affects the acquisition of mCAs. We instead focused the current study on studying the mechanisms of clonal selection shaped by the interaction between mCAs and gene mutations, as well as their conjoint role in evolution from CH to leukemia. Future studies with prospective collection of longitudinal samples are warranted to resolve the role of therapy.

4) Following up on this, could it be that presence of mutational changes in peripheral blood before onset of therapy is an indication of the overall mutational burden of organism and might predict overall susceptibility to leukemia development.

We thank the reviewer for raising this interesting hypothesis. It is indeed possible that CH mutations that pre-exist before therapy play an important role in shaping leukemia risk. For example, it has been shown that *TP53* mutations can be present at low allele frequencies before the start of treatment and preferentially expand upon cytotoxic therapy to drive the subsequent therapy-related myeloid neoplasm (t-MN) (Wong et al, Nature 2015). In a separate study, we have further demonstrated that CH mutations in the DDR pathway (*TP53*, *PPM1D*, *CHEK2*) have a stronger fitness advantage under exposure to certain classes of oncological therapy and are a significant factor in determining subsequent t-MN risk (Bolton et al, in press, doi: <https://doi.org/10.1101/848739>). However, the study of pre-existing CH mutations before onset of

therapy requires sensitive detection of deeply subclonal mutations, and given the nature of our data (targeted panel sequencing) we have limited capability to detect mCAs at low cell fractions (<10%). In addition, we have limited capacity to model the effect of therapy since complete treatment history is only available for a subset of patients (10,375 out of 32,444). We therefore hope the reviewer agrees that this would be out of scope of the current study, and future studies with more sensitive mCA detection methods and more complete treatment annotations are warranted to evaluate this hypothesis.

5) The follow up for blood diseases in this cohort is rather very short and should be at least 5-10 years and any . The authors report onset of multiple myeloma. Multiple myeloma always arises of monoclonal gammopathy or smouldering myeloma and this precedes the onset of multiple myeloma several years. So the condition might have been present before. The same also applies for NHL. May be this part should be discusses in depth with the other coauthors who are excellent clinician scientists

The reviewer is correct that the follow-up of our study is relatively short for blood diseases. This reduces our power to determine long-term effects of CH on the development of leukemia and our capacity to estimate the risk of specific events (e.g. individual chromosomal alterations and interactions with specific gene mutations). However, our current length of follow-up is sufficient to identify the co-occurrence of mCAs and gene mutations as a significant risk factor for leukemia transformation within a 3-4 year period (Fig. 4c,d). We believe the results from this study would serve as the foundation for designing future studies with longer follow-up periods to fully determine the long-term effects of mCAs and specific co-mutations. We have added the following sentence to the discussion to address this limitation:

“It is also of note that due to the relatively short follow-up, we are limited in our capacity in assessing the long-term effects of mCAs on leukemia risk, which would be a goal for future studies.”

With regards to the onset of multiple myeloma and NHL, we agree with the reviewer that due to the chronic nature of these conditions, they could have existed in parallel at the time of CH assessment. In addition, it is unclear whether the CH mutations observed prior to diagnosis are related to the resulting tumor (Fig. 4a). For this reason we decided to exclude multiple myeloma and NHL from subsequent analysis and instead focus on leukemias. We apologize if this was not clear. We have modified the below sentence in the results section “Evolution to hematologic malignancies” to provide further clarifications:

“It is unclear whether these mutations were related to the subsequent diagnosis and we therefore restricted our subsequent analysis on the risk of leukemias.”

6) The method for follow up for blood disease using billing information can be associated with high degree of bias (patients might not be able to afford additional treatment, might go another center) and is rather short.

The reviewer is right that patients can be lost to follow-up due to a variety of reasons, and we are unable to capture leukemia diagnosed later at a different center after patients leave MSKCC. In our risk modeling using cause-specific Cox regression analysis, we take this into account by treating patients who were lost to follow-up as right-censored data points (the other two endpoints are leukemia diagnosis and death), and we do not include them in the risk set at subsequent timepoints. In addition, loss of follow-up can be due to a variety of additional reasons such as moving locations and change of insurance. We therefore think that loss of follow-up is unlikely to be a source of major bias.

7) My biggest concern regards Extended data figure 6:

It is really interesting that beam irradiation shall increase onset of mCA and cytotoxic therapy shall reduce this risk (Odds ratio smaller than 1, albeit with a high CI).

This just can not be. Irraditaion therapies are really diverse but it has been shown in many studies, that chemotherapy is the one force driving leukemia development. With this results, I do not know whether there is a strong cofounde bias for selection of patients and interpretation of data.

We appreciate the fact that chemotherapy plays an important role in the development of secondary leukemia (McNerney et al, Nature Reviews Cancer 2017; Coombs et al, Cell Stem Cell 2017). In fact, we investigated the relationship between oncologic therapy, CH gene mutations, and their combined impact on therapy-related myeloid neoplasms (t-MNs) in a subset of our current cohort in a separate study (Bolton et al, in press, doi:<https://doi.org/10.1101/848739>). We found that radiation and specific subclasses of cytotoxic therapy promote the clonal expansion of pre-existing CH mutations in genes involved in the DDR pathway (*TP53*, *PPM1D*, *CHEK2*) which in some patients went on to seed the corresponding t-MN. This observation led us to investigate the potential impact of therapy on mCAs.

We would like to note that our regression result (HR=0.88, p=0.56, 95% CI=0.57-1.36) does not suggest that cytotoxic therapy reduces the risk of mCAs. As the reviewer has correctly pointed out, the confidence interval is wide due to our small sample size (113 mCAs total in 10,375 individuals with treatment information). The true effect size can therefore be estimated to be anywhere between 0.57 and 1.36, which includes both negative and positive regions of the potential effect. In fact, a previous work by Jacobs et al investigated the effect of therapy on mCA risk in a larger cohort (n=31,717), which also did not find a significant positive association of therapy and mCA. These results are not inconsistent with the fact that chemotherapy is a major driver of leukemia development, as t-MN pathogenesis is a multifactorial process and could involve mutation types other than chromosomal alterations, such as structural variants and gene mutations (McNerney et al, Nature Reviews Cancer 2017; Bolton et al, in press, doi:<https://doi.org/10.1101/848739>).

In addition, as the reviewer pointed out, oncologic therapies are very diverse and their impacts on CH risk likely depend on the specific drug classes received as well as the specific CH alterations assessed. It is certainly possible that specific subclasses of chemotherapy are associated with specific types of mCA, as it is the case for CH gene mutations (Bolton et al, in press), and these associations would not be detected in a pooled analysis. However, due to the rarity of detectable

mCAs (n=383) as compared to gene mutations (n=14,789) we lack the power to assess the effects of specific subclasses of chemotherapy on the risk of specific mCA alterations.

To address the reviewer's concern about confounding and selection bias, we have ascertained that with our current dataset, significant positive associations of gene mutations with prior receipt of both cytotoxic therapy (OR=1.29, $p=1.6 \times 10^{-7}$) and XRT (OR=1.64, $p=1.2 \times 10^{-22}$) can be recapitulated. We therefore find it unlikely that we did not find a positive association of mCAs with cytotoxic therapy due to a strong confounding bias.

To summarize, our result does not suggest that cytotoxic therapy influences the risk of mCAs in a negative or positive direction. We would like to think of this result as a hypothesis generation step for future studies with larger cohorts and more detailed clinical annotations (broken down into drug classes) and/or longitudinal samples before/after therapy, which could more systematically answer the question regarding the potential effect of therapy on the risk of mCAs. To make this interpretation more clear and avoid confusion, we have modified this sentence in the results section "Landscape of mCAs in prospective sequencing of cancer patients":

"However, the association between therapies and mCAs is likely to be heterogeneous²⁶. Larger datasets with more complete treatment annotation are needed to fully assess the association between specific subclasses of therapy on specific types of mCAs."

8) The fact that cancer therapy predisposes to leukemia development is not really novel. The authors report an interesting clonal evolution but what would be the really new information? The impact of different therapies on clonal evolution would be interesting but this would be needed to demonstrate in more detail

We agree with the reviewer that cancer therapy confers higher risk of leukemia is not a novel observation. Given our limited capacity to capture the impact of diverse classes of therapies, we focused our investigation on the relationship between mCA and gene mutations in CH and their conjoint role in shaping clonal evolution from CH to leukemia. This analysis has generated several novel biological and clinical insights that could not be captured with either mutation class alone. First, our study is the first to map mCAs together with gene mutations in a large cancer population. Second, although mCAs have been demonstrated to be associated with leukemia (Jacobs et al, Nature Genetics 2012; Laurie et al, Nature Genetics 2012; Loh et al, Nature 2018; Loh et al, Nature 2020), a new finding of our study is that mCAs with concurrent gene mutations delineates a subset of cancer patients with especially high risk of developing leukemia (3-year cumulative incidence=14.6%, CI:7-22%) compared to patients with only mCAs, only gene mutations, or no CH (all of which had 3 year CI of less than 1%). Last, to the best of our knowledge, it has not been reported that mCA is an independent risk factor for subsequent leukemia development apart from gene mutations. Our multivariate model integrating features of gene mutations and mCAs revealed that mCA is independently predictive of subsequent leukemia development (HR=14, 95% CI:6-33, $P<0.001$). Taken together our study is the first to provide a detailed evaluation of the relationships between mCAs and acquired gene mutations in early leukemogenesis and the evolution of CH. In

addition, we develop and provide an open access method that enables mapping of mCAs from clinical panel sequencing assays, and propose a framework on how CH assessment can be incorporated into the development of future clinical decision support algorithms for risk stratification, surveillance and early detection.

Given these new insights on the interplay between mCAs and gene mutations in CH, it would indeed be interesting to investigate the role of different classes of therapy in shaping the patterns of clonal evolution. However, as discussed above, due to the rarity of detectable mCAs (n=383) as compared to gene mutations (n=14,789) as well as the fact that we only have treatment annotation for a subset of patients (10,375 out of 33,442) our dataset is not well-suited to address this question. We would like to clarify that the dataset is well-powered to explore the evolutionary relationships between acquired gene mutations and mCAs and how their interaction relates to the risk of subsequent leukemia, which is why we opted to focus our manuscript in this direction and limited analysis with regards to treatment in the supplementary information. We hope the reviewer would agree that this would be out of scope of the current study and future studies with larger cohorts and more complete clinical annotations of different drug classes and/or longitudinal samples during therapy are warranted to fully answer this question.

Or mutational burden before onset of therapy as an indication of potential overall mutational burden would be an interesting point.

As discussed in response to the reviewer's fourth question, while this is an interesting point to investigate indeed, the study of pre-existing CH mutations before onset of therapy would require more sensitive detection methods for deeply subclonal mCAs (cell fraction <10%) and more complete treatment annotations. We therefore hope the reviewer would agree that this would be out of scope of the current study and would be a direction for future research.

The association of certain mCA and different hematological diseases would be interesting.

We thank the reviewer for the opportunity to address this insightful point. We indeed observe highly specific types of CH mutations characteristic of the respective disease types for patients who subsequently developed MDS and MPN (e.g. *EZH2*/7qCNLOH in MDS, *JAK2*/9pLOH in MPN). We present this data in Fig. 4a and discuss it in the results section "Evolution to hematologic malignancies". However, the number of mCA events (n=383) and the number of patients who subsequently developed leukemia (n=60) in this study is too low to conduct a rigorous analysis on specific chromosomal alterations and this has been thoroughly investigated by prior studies in bigger datasets that only mapped mCAs (Loh et al, Nature 2018; Loh et al, Nature 2020). We therefore primarily focused on the opportunity to characterize the interaction between gene mutations and mCAs, and how their interplay shapes the evolution from CH to leukemia.

REVIEWERS' COMMENTS

Reviewer #1 (Remarks to the Author):

No further comments. Congratulations to the authors

Reviewer #2 (Remarks to the Author):

The authors have addressed my comments and I am happy with the revised manuscript. I congratulate the authors again on a very nice study.

Reviewer #3 (Remarks to the Author):

Although the authors wrote a lengthy point by point answer, they did not answer most of my concern but stated instead that these concern were not part of the study but more elaborate studies have to be undertaken. I fully accept the high quality of the techniques applied.

There is room for discussion whether or not to accept the publication but I still have major concern regarding the clinical claims of this manuscript.

Especially one figure reflects my concern regarding the highly controversial claims of the paper:

I already pointed my concern concerning extended figure six.

The authors did not respond to my concern regarding inserting any reference to cytotoxic or beam therapy. This simply can not be true or there is a huge confounding error. The fact that this has emerged as part of the discussion and the fact that the authors refuse to take this out leads to big concern from my side, that a number of additional claims could be wrong. This becomes dangerous, if clinical investigation or (hopefully not) decisions are at least partially based on this manuscript.

RESPONSE TO REVIEWERS

Reviewer #1 (Remarks to the Author):

No further comments. Congratulations to the authors

We thank the reviewer for the positive assessment and insightful comments that have helped us improve the manuscript.

Reviewer #2 (Remarks to the Author):

The authors have addressed my comments and I am happy with the revised manuscript. I congratulate the authors again on a very nice study.

We thank the reviewer for the positive assessment and insightful comments that have helped us improve the manuscript.

Reviewer #3 (Remarks to the Author):

Although the authors wrote a lengthy point by point answer, they did not answer most of my concern but stated instead that these concern were not part of the study but more elaborate studies have to be undertaken. I fully accept the high quality of the techniques applied.

There is room for discussion whether or not to accept the publication but I still have major concern regarding the clinical claims of this manuscript.

Especially one figure reflects my concern regarding the highly controversial claims of the paper:

I already pointed my concern concerning extended figure six.

The authors did not respond to my concern regarding inserting any reference to cytotoxic or beam therapy. This simply can not be true or there is a huge confounding error. The fact that this has emerged as part of the discussion and the fact that the authors refuse to take this out leads to big concern from my side, that a number of additional claims could be wrong. This becomes dangerous, if clinical investigation or (hopefully not) decisions are at least partially based on this manuscript.

We apologize that our response did not address the reviewer's concern. The results in the regression presented in Extended Data Figure 6 (now Supplementary Figure 10) do not suggest that cytotoxic therapy reduces the risk of mCAs (HR=0.88, p=0.56, 95% CI=0.57-1.36), and we do not make that claim in the manuscript. The correct interpretation is that cytotoxic therapy in the present dataset does not show an association with mCAs. Importantly, this is in agreement with

prior literature (Jacobs et al 2012 Nature Genetics, PMID: 22561519). Notably, a similar frequency of mCAs was observed in previously treated patients as compared to a cancer-free population in Jacobs et al.

It is also important to highlight that our study is observational and as no intervention was performed, the analysis presented in Supplementary Figure 10 is not intended to uncover causal effects of therapy exposures on mCAs or serve as the basis for any clinical trial. To clarify this point, we have added exact *P* values in Supplementary Figure 10 and added the below sentence to the figure legend: “The associations presented in this figure do not represent causal relationships”.

With regards to the reviewer’s comment about potential confounding bias in the dataset, we have recently published a study investigating the detailed relationships between oncologic therapy and CH as defined by gene mutations (gmCH) in the same dataset (Bolton et al, Nature Genetics 2020). In this study we derive some important insights on the patterns of selection and association between molecular subtypes of CH and cancer therapy. These include:

1. gmCH is indeed enriched in patients that had received cytotoxic and external beam radiation therapy prior to blood draw.
2. gmCH mutations targeting genes in the DNA Damage Response (DDR) pathway such as *TP53*, *PPM1D* and *CHEK2* are specifically enriched in previously treated patients as opposed to those who are treatment naive. This suggests that the effects of cancer therapy on CH vary by mutation type. In Bolton et al, we further validate this hypothesis through sequential sampling where we show that therapy selects for clones with CH in DDR genes but not others (e.g. *DNMT3A*, *TET2*, *ASXL1*).
3. Evidence of differential effects among distinct types of therapeutic agents. We characterize >10 therapeutic agents and show that not all classes of oncologic therapy have the same effects and that any observed associations are further underlain by agent-specific effects. For example, carboplatin but not oxaliplatin agents underlie most of the associations between CH and platinum therapy.

Taken together our findings demonstrate that this dataset is appropriately annotated to characterize the relationship between cancer therapy and CH. In the present manuscript the observation that we detect an association of mCA with external beam radiation therapy but not cytotoxic therapy supports the hypothesis that the effect of therapy on mCAs is agent-specific. However, given (1) mCAs detectable by our sequencing panel are markedly less frequent (~1% of the cohort) than gmCH (~33% of the cohort) and (2) the diverse mechanisms underlying the relationship between CH and distinct classes of cytotoxic therapies, the power to detect and quantify such associations is limited with our current cohort size.

Thus, we respectfully maintain that these are important results to present and that indeed larger datasets and future analyses are warranted to further our understanding of the diverse mechanisms of mCA selection and the effect of cancer therapy on the evolution of CH, which is an emerging and very important area of research.

Importantly, the central focus of the manuscript involves selection mechanisms of mCA through cooperation with CH gene mutations, accounted for by *cis*-effects (allelic dosage adjustment of gene mutations in cis of mCA) and epistatic interactions. Thus, consideration of the study findings as a whole should balance the limitation in delineating the effects of cancer therapy.

To address the reviewers concern we have also embedded the following paragraph in the discussion:

“We recapitulate known associations of mCA with age (OR=1.8, P<0.001), male gender (OR=1.4, P=0.012) and white race (OR=1.5, P=0.033)¹⁴⁻¹⁶. We find that mCAs are significantly associated with external beam radiation (OR=1.7, P=0.022) but not cytotoxic chemotherapy (OR=0.9, P=0.56). This result is in line with the highly heterogeneous effects of cancer therapy on CH shown in recent studies^{25,26}. It is possible that associations may exist between specific types of mCAs and specific classes of cytotoxic therapy, but delineation of such effects will require larger datasets with more detailed treatment annotations.”